# Pandemic Analytics by Advanced Machine Learning for Improved Decision Making of COVID-19 Crisis

**Konstantinos Demertzis [1,2,*]**, **Dimitrios Taketzis [3]**, **Dimitrios Tsiotas [4,5]**, **Lykourgos Magafas [1]**, **Lazaros Iliadis [2]** and **Panayotis Kikiras [6]**

1    Laboratory of Complex Systems, Department of Physics, Faculty of Sciences, Kavala Campus, International Hellenic University, 65404 St. Loukas, Greece; magafas@teikav.edu.gr
2    Faculty of Mathematics Programming and General Courses, Department of Civil Engineering, School of Engineering, Democritus University of Thrace, Kimmeria, 67100 Xanthi, Greece; liliadis@civil.duth.gr
3    Hellenic National Defence General Staff, Stratopedo Papagou, Mesogeion 227-231, 15561 Athens, Greece; d.taketzis@hndgs.mil.gr
4    Department of Regional and Economic Development, Agricultural University of Athens, Nea Poli, 33100 Amfissa, Greece; tsiotas@aua.gr
5    Department of Planning and Regional Development, University of Thessaly, Pedion Areos, 38334 Volos, Greece
6    School of Science, Department of Computer Science, University of Thessaly, Galaneika, 35131 Lamia, Greece; kikirasp@uth.gr
*    Correspondence: kdemertzis@teiemt.gr

**Abstract:** With the advent of the first pandemic wave of Severe Acute Respiratory Syndrome Coronavirus-2 (SARS-CoV-2), the question arises as to whether the spread of the virus will be controlled by the application of preventive measures or will follow a different course, regardless of the pattern of spread already recorded. These conditions caused by the unprecedented pandemic have highlighted the importance of reliable data from official sources, their complete recording and analysis, and accurate investigation of epidemiological indicators in almost real time. There is an ongoing research demand for reliable and effective modeling of the disease but also the formulation of substantiated views to make optimal decisions for the design of preventive or repressive measures by those responsible for the implementation of policy in favor of the protection of public health. The main objective of the study is to present an innovative data-analysis system of COVID-19 disease progression in Greece and her border countries by real-time statistics about the epidemiological indicators. This system utilizes visualized data produced by an automated information system developed during the study, which is based on the analysis of large pandemic-related datasets, making extensive use of advanced machine learning methods. Finally, the aim is to support with up-to-date technological means optimal decisions in almost real time as well as the development of medium-term forecast of disease progression, thus assisting the competent bodies in taking appropriate measures for the effective management of the available health resources.

**Keywords:** COVID-19; pandemic; data analytics; prediction; decision making; machine learning

## 1. Introduction

The health crisis caused by the SARS-CoV-2 pandemic, combined with the economic consequences and the shock to health systems, has created serious concerns on how to make timely and valid decisions about prevention and social distancing measures to be taken [1]. The COVID-19 pandemic has created a rapidly changing environment where a huge amount of data related to virus spread updates is daily presented. The effective utilization of this data and the provision of the thorough and at the same time fast analysis of the most up-to-date information to support the best decisions requires their intelligent processing in near real time [2].

The analysis of the spread rate of COVID-19 disease is directly related to the general concerns and challenges of large-scale near real-time data analysis procedures. Specifically, it is directly related to the high velocity with which the relevant information arrives, how this information is collected and stored (its volume), the variety of unstructured or semi-structured data forms that can be collected, their variability as epidemiological data change in importance over time, their visualization and the diagnosis of whether the information is accurate or incomplete and inaccurate (its veracity), and finally determining their final operational value [3]. Understanding how the parameters of these data are linked can help civil protection organizations identify in a clear and fully understandable way what capabilities they need to develop or acquire to make full use of the data they have to strengthen public safety, health, and consequently safeguarding the state's health system [4].

Beyond their management, the biggest modern challenge for large-scale data such as those related to COVID-19 disease is to analyze them functionally to finally reveal the hidden knowledge contained in this information. For example, using pattern recognition methods, it is possible to identify trends or patterns, to identify unknown correlations, as well as other useful information, to achieve behavioral prediction and make optimal decisions [5]. It is important to note that the above analysis can be used not only to implement appropriate policies to prevent and deal with future epidemics by giving a retrospective picture of the pace and ways of its spread but also to make optimal decisions and actions in almost real time [6].

This very ability to process huge amounts of data, using advanced algorithms and generally intelligent analysis and processing tools, is a very promising solution to the effective detection and tracing of active cases, while also creating the background for the development of spatio-temporal solutions adapted to real needs, but also methods of timely forecasting of potential threats to public health [7].

Due to the extremely urgent need to take action to reduce the spread of the disease, the requirements of civil and health protection mechanisms must include appropriate algorithms for fast to instantaneous processing of large volumes of data with high complexity, and possible high inhomogeneity [8]. In general, the approaches that should be chosen to shield the public health system should meet specific specifications, ensuring at least multiple design aspects, such as [9]:

1. Integrated and interoperable data representation.
2. Intelligent data management methods (time-series analysis, anomaly detection, dimensional reduction, parameter selection, etc.).
3. Real-time analysis mechanisms.
4. Ability to securely exchange data between distributed systems.

The above requirements have led to the parallel development of both the infrastructure that supports large-scale data and the algorithmic standardizations that must be followed to ensure public health [10]. In this spirit, the study of how to record, analyze, and model the problem of the spread of the disease is extremely important, both from an epidemiological point of view and from a mathematical point of view [11].

This paper proposes a novel model for the near-real-time analysis of COVID-19 disease data, as well as an intelligent machine learning system for predicting disease progression, in order to assist in deciding on predictive or suppressive measures of social distancing or taking appropriate measures related to the management of the health system. The proposed system is based on automated data collection and analysis, while the medium-term forecast is based on advanced machine learning methods. Within this context, the proposed method can be applied to different aspects of the COVID-19 temporal spread in Greece and her border countries to present an exploratory study of COVID-19 disease progression (real-time statistics about the cumulative number of infections, deaths, ICU patients, and epidemiological indicators). In practical implementation, the proposed methodology offers an active method for modeling and forecasting the pandemic, which is capable of removing the disconnected past data from the time-series structure in order to provide a modeling and

forecasting tool facilitating decision making and resource management in epidemiology, which can contribute to the ongoing fight against the pandemic of COVID-19.

The rest of the work is structured as follows. Initially, relevant research papers are presented on how to record, analyze, and model the problem of pandemic spread. Then, the third section presents the way of mathematical modeling and analysis of epidemiological data using non-spatial causal models and indicators. The time-series forecasting methodology is presented in the next section, while chapter five presents the data used and the results obtained. Finally, in the last section, there is an extensive analysis and discussion of the general methodology that took place, and the study closes with the presentation of future research that is proposed to be followed.

## 2. Related Work

Methodologies for mathematical modeling of the spread of the disease [12] and especially techniques for predicting the future variation of the epidemic curve [13] are deemed as a constant demand by the research community, with remarkable findings already recorded, offering an important legacy of knowledge [14–16].

For example, the detailed research of Sarkodie et al. [17] temporally models the evolution of the pandemic, constructing at the same time conceptual tools for linking the relationships between confirmed cases and deaths, based on four characteristic health indicators. The final assessment of this research is based on cross-sectional dependence, endogeneity, and unobserved heterogeneity. Although the linear relationship between deaths and confirmed cases are revealed, as well as the non-linear correlation between recovery cases and confirmed cases, the study fails to provide a final model with substantial generalization possibilities as it uses limited in scale non-critical data that cannot be used for extensive identification of the phenomenon.

On the other hand, the purpose of this work [18] is to give a contribution to the understanding of the COVID-19 contagion in Italy. To this end, the authors developed a modified Susceptible–Infected–Recovered–Deceased (SIRD) model for the contagion, and they used official data of the pandemic for identifying the parameters of this model. Their approach features two main non-standard aspects. The first one is that model parameters can be time-varying, allowing them to capture possible changes of the epidemic behavior, due for example to containment measures enforced by authorities or modifications of the epidemic characteristics and to the effect of advanced antiviral treatments. The time-varying parameters are written as linear combinations of basis functions and are then inferred from data using sparse identification techniques. The second non-standard aspect resides in the fact that they consider as model parameters also the initial number of susceptible individuals, as well as the proportionality factor relating the detected number of positives with the actual (and unknown) number of infected individuals. Identifying the model parameters amounts to a non-convex identification problem that they solve by means of a nested approach, consisting of a one-dimensional grid search in the outer loop, with a Lasso optimization problem in the inner step.

In contrast, Anastassopoulou et al. [19], using more complete datasets and heuristic methodology for estimating epidemiological parameters, model the rates of disease spread with a much more complete and substantial contribution to the way the pandemic is assessed. However, the reverse prediction process based on spread scenarios, which reproduces the confirmed hypotheses, creates a directed trend that is part of a very specific framework, suitable only for the verification of simulation techniques.

A fully technical prototype of high research interest was presented in the work of Fong et al. [20], where they presented an optimized prediction model of polynomial neural networks with corrective feedback, which can generalize, even in cases where the samples are minimal. Although the methodology is very robust, it needs to be compared with competing algorithms, taking into account additional process evaluation criteria apart from those describing the level of accuracy/error.

Differently from the related literature, where modeling and controlling the pandemic contagion is typically addressed on a national basis, this paper [21] proposes an optimal control approach that supports governments in defining the most effective strategies to be adopted during post-lockdown mitigation phases in a multi-region scenario. Based on the joint use of a non-linear Model Predictive Control scheme and a modified Susceptible–Infected–Recovered (SIR)-based epidemiological model, the approach is aimed at minimizing the cost of the so-called non-pharmaceutical interventions (that is, mitigation strategies), while ensuring that the capacity of the network of regional healthcare systems is not violated. In addition, the proposed approach supports policymakers in taking targeted intervention decisions on different regions by an integrated and structured model, thus both respecting the specific regional health systems characteristics and improving the system-wide performance by avoiding uncoordinated actions of the regions. The methodology is tested on the COVID-19 outbreak data related to the network of Italian regions, showing its effectiveness in properly supporting the definition of effective regional strategies for managing the COVID-19 diffusion.

Given the scale of the pandemic in different countries, many researchers have focused on local analyses based on officially available data. For example, Mahase et al. [22] present the statistical data of the United Kingdom after the implementation of social distancing. A particularly detailed research effort to localize the phenomenon is presented in the article [23], which explores the spatio-temporal trend of the epidemic in Italy. This study is based solely on statistical modeling without taking into account the statistical significance tests used to test the scientific hypothesis that is initially taken into account. The severity of this weakness is magnified by the fact that the object of epidemiological studies is an occurrence function and more specifically a measure of association that quantifies the relationship between the identifier studied and the outcome, which is required to decide whether this relationship is statistically significant or not.

Respectively, focusing on the peculiarities of the spread of COVID-19 in Greece, ref. [12] offers an exploratory time study of the course of the disease while at the same time proposing a realistic model for predicting high reliability. Specifically, a statistical analysis of the evolution of epidemiological data in Greece is presented, where the rate of spread and the perceived spread of the disease are approximated and standardized with mathematical standards. Respectively, a methodology for predicting the high solvency of total cases, deaths, and intensive care unit beds is proposed based on the Regression Splines algorithm. The important innovation of the proposed model is that it bases its operation on the previous modeling with a Complex Network of the social distancing measures taken in Greece, thus implementing a fully functional and realistic system of evaluation and interpretation of disease-related events.

Evolving the above investigation, ref. [13] attempts to anticipate the "Flattening of the Curve", to make optimal decisions regarding the support of the health system and the implementation of additional measures being taken, such as a reduction of social distancing. The proposed system approaches offer realism in the way of their evaluation while offering a powerful mechanism for modeling the spread of the pandemic.

The local evaluation of the phenomenon, while it is an essential basis of evaluation, also contains serious weaknesses if it is not based on solid conditions. For example, a subjective approach in predicting disease spread based on exponential smoothing models is presented in the paper [24]; here, the trend index, which is calculated following the pattern of the disease of the past based on local data and the smoothing of the curve, is predicted based on similar case studies of other countries leading the pandemic.

Focusing on the specifics of the spread of the disease both epidemiologically and in terms of the implementation of preventive and repressive measures, this paper presents an exploratory study for the near real-time analysis of large-scale disease data with advanced intelligent machine learning techniques, which uses the visualized material that can be produced by the corresponding information system. The aim is to reveal the knowledge hidden in the epidemiological data, deciphering, and capturing the mathematics of the

pandemic and specifically the indicators that can model the spatio-temporal evolution and the spread of the disease.

## 3. Mathematical Modeling and Pandemic Analytics

Spatio-temporal modeling of the circulation of pathogens between hosts and through transmitters is used to simplify the reality or complex correlations associated with a chaotic phenomenon such as the pathogen–host interaction [25]. In particular, mathematical modeling, especially when performed in real time, is a powerful tool for studying the dynamic transmission of infectious diseases using non-spatial causal models (Susceptible Infectious, Recovered—SIR) and in general in assisting in optimal decision making [26].

Decision making in epidemiology [27] is based on predicting or simulating behaviors and properties of complex systems based on mathematical modeling. Epidemiology is the study of the distribution and evolution of various diseases in the human population (descriptive epidemiology) and the factors that shape them or can influence them (analytical epidemiology) [28].

### 3.1. Real-Time Statistics

Greece at the time of completing the study (17 June 2021) had 417,253 coronavirus cases, 12,488 deaths, and 396,317 recovered, with daily variance as shown in Figure 1 [29].

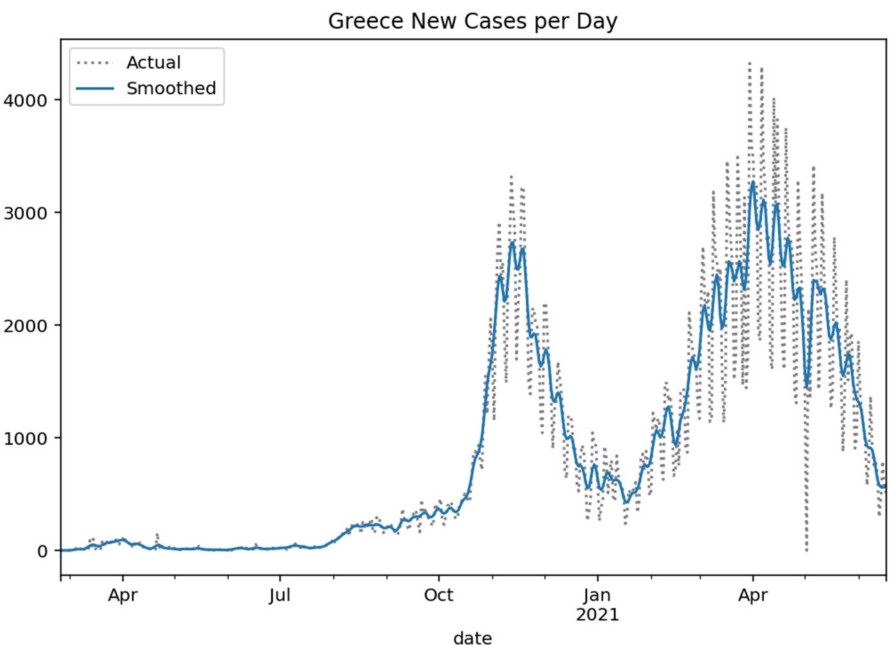

**Figure 1.** Greece new cases per day (Gaussian smoothed).

Respectively, the following Figures 2–5 show the daily variation of the cases with Greece's neighboring countries (Albania, Bulgaria, Turkey, and North Macedonia) to assist the decision-making system and the corresponding social distancing mechanisms [29].

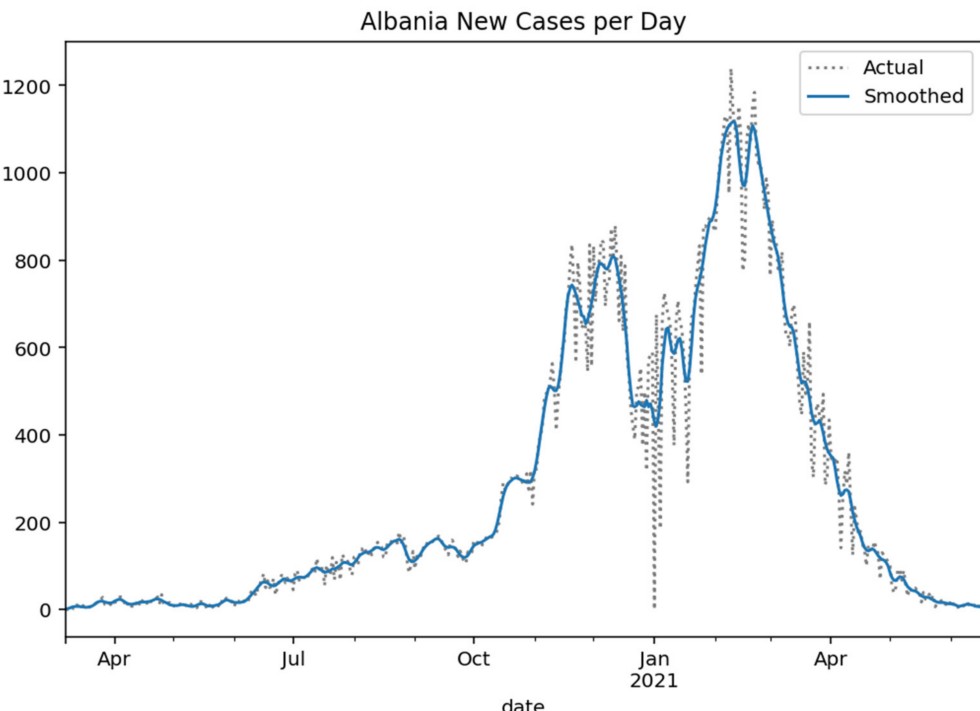

**Figure 2.** Albania new cases per day (Gaussian smoothed).

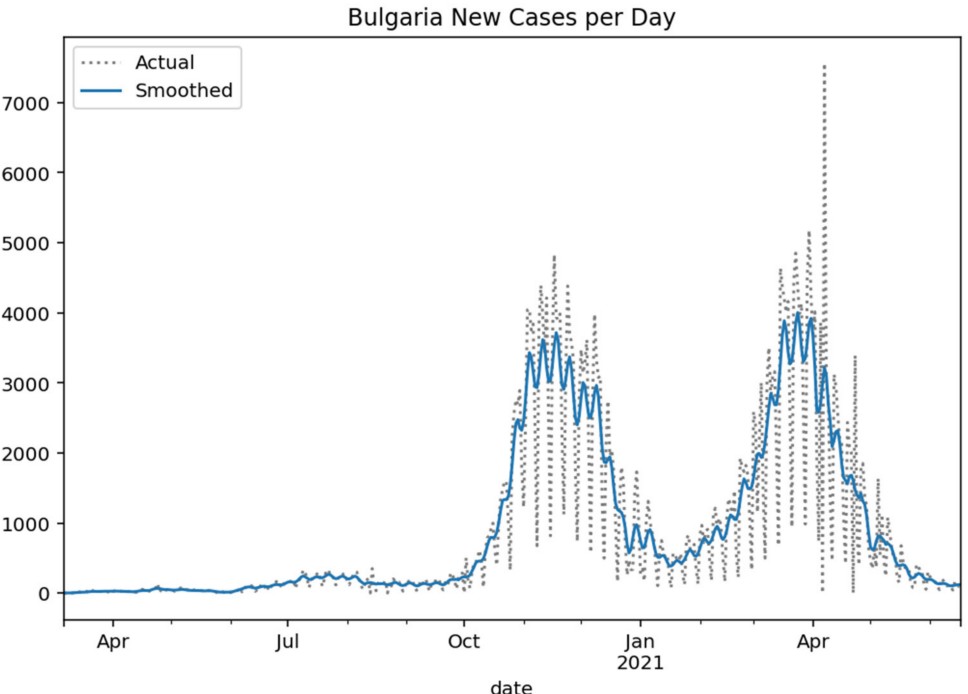

**Figure 3.** Bulgaria new cases per day (Gaussian smoothed).

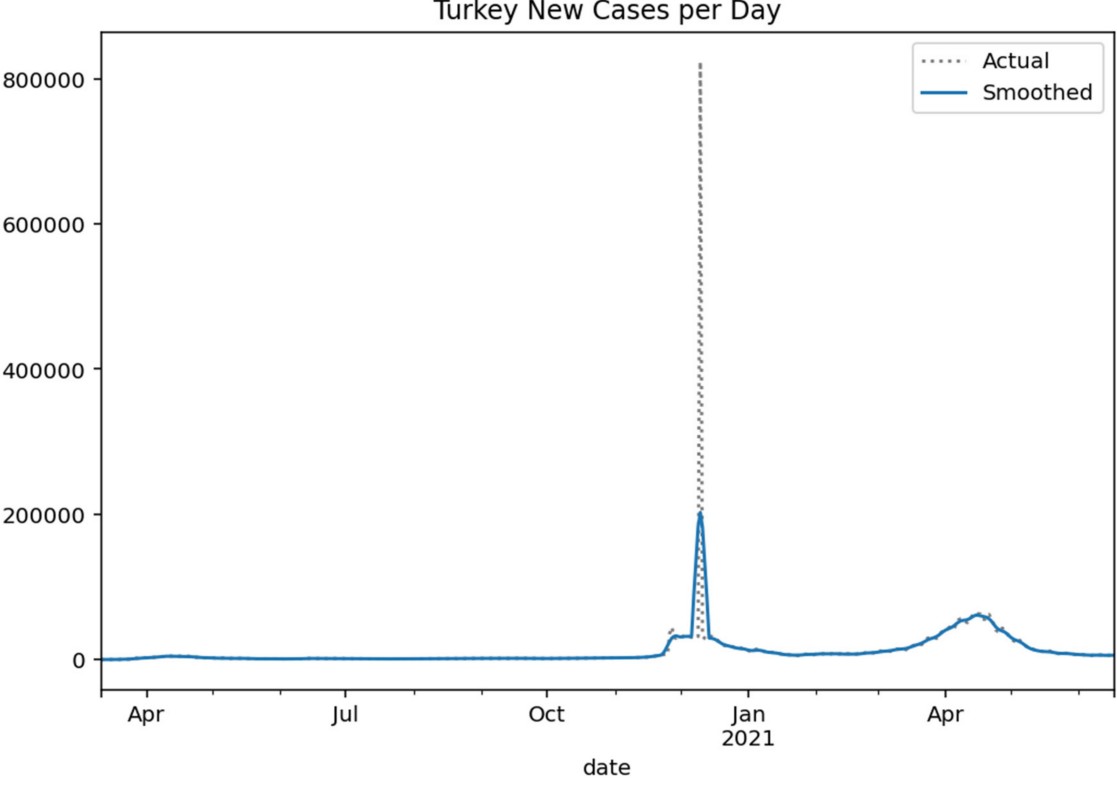

**Figure 4.** Turkey new cases per day (Gaussian smoothed).

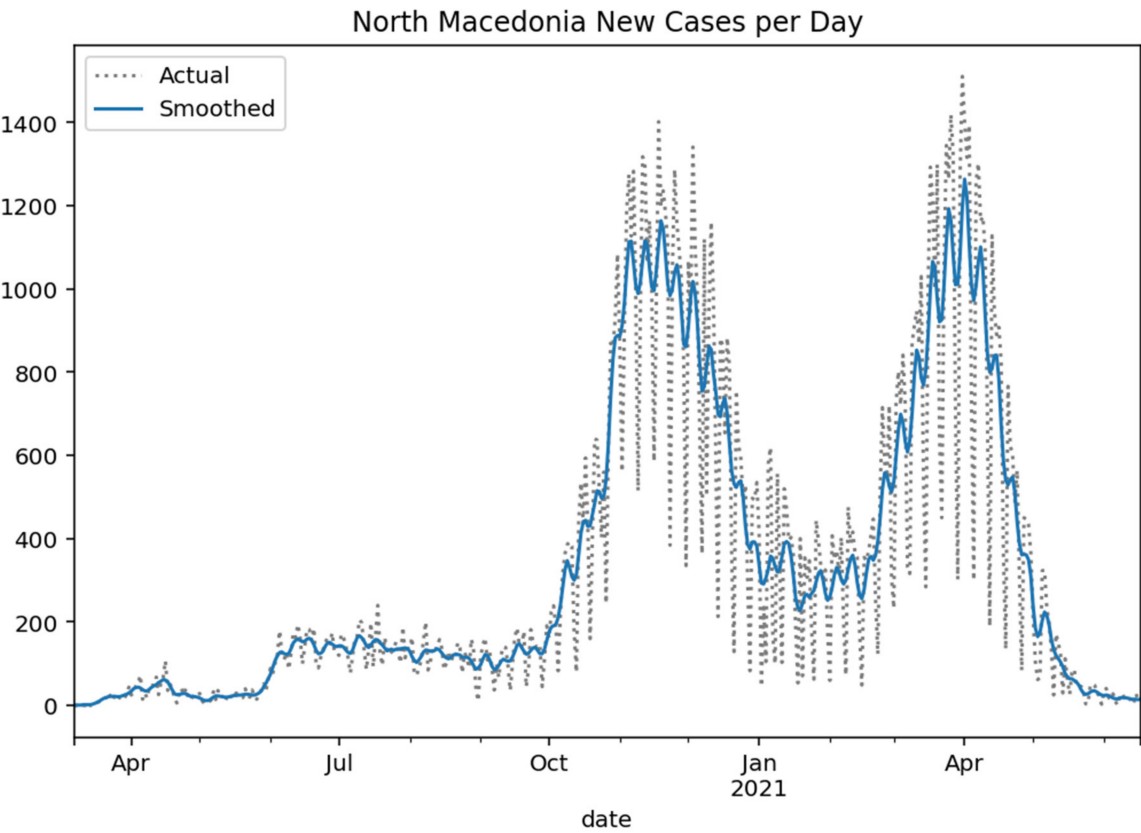

**Figure 5.** North Macedonia new cases per day (Gaussian smoothed).

For the most complete and effective decision making, real-time statistical analysis of the pandemic is required at a level where the technical characteristics of the problem can be captured. Detailed statistical analysis for Greece is presented in the following Tables 1–4 [29]:

**Table 1.** Pandemic Statistic Analysis_1.

|  | Total Cases | New Cases | Total Deaths | Reproduction Rate | Weekly ICU Admissions |
|---|---|---|---|---|---|
| mean | 109,424.8912 | 872.9142259 | 3475.760776 | 1.074684096 | 112.8019394 |
| std | 131,362.2468 | 1008.649671 | 4084.12127 | 0.21673656 | 119.1627889 |
| min | 1 | 0 | 1 | 0.69 | 1.945 |
| max | 417,253 | 4322 | 12,488 | 1.58 | 382.165 |

**Table 2.** Pandemic Statistic Analysis_2.

|  | New Tests | Total Tests | Total Tests/1000 | New Tests/1000 | Positive Rate | Tests Per Case |
|---|---|---|---|---|---|---|
| mean | 23,151.10644 | 3,056,182.995 | 293.2137222 | 2.221138614 | 0.034083871 | 84.47204301 |
| std | 21,782.07213 | 3,037,565.947 | 291.4275674 | 2.089800416 | 0.025939906 | 122.2038529 |
| min | 45,335 | 570 | 0.055 | −4.349 | 0.001 | 9.5 |
| max | 130,207 | 10,207,626 | 979.331 | 12.492 | 0.105 | 768.2 |

**Table 3.** Pandemic Statistic Analysis_3.

|  | Total Vaccinations | People Vaccinated | People Fully Vaccinated | New Vaccinations | Total Vaccinations/1000 |
|---|---|---|---|---|---|
| mean | 220,830,6.81 | 1,455,548.608 | 887,829.8134 | 47,651.25564 | 21.18607843 |
| std | 212,368,4.319 | 1,346,725.121 | 826,843.6484 | 36,188.52581 | 20.37470194 |
| min | 447 | 447 | 2 | 147 | 0 |
| max | 7,244,517 | 4,381,177 | 3,045,889 | 114,676 | 69.5 |

**Table 4.** Pandemic Statistic Analysis_4.

|  | People Vaccinated/1000 | People Fully Vaccinated/1000 | Stringency Index | Hospital Beds/1000 | % Death/Cases |
|---|---|---|---|---|---|
| mean | 13.96457516 | 8.518432836 | 68.39331197 | 4.21 | 3.385838864 |
| std | 12.92053129 | 7.932278859 | 16.53239424 | $2.49 \times 10^{-14}$ | 1.412956795 |
| min | 0 | 0 | 11.11 | 4.21 | 0 |
| max | 42.03 | 29.22 | 88.89 | 4.21 | 6.134338588 |

It should be noted that the stringency index is an index provided by the Oxford COVID-19 Government Response Tracker [30], which includes a team of one hundred experts, who constantly update a database with 17 government response indicators, considering restraint policies such as school and workplace closures, public events, public transportation, home accommodation policies, etc. Essentially, it is a number ranging from 0 to 100 that reflects the 17 rating indicators, with the highest score indicating the highest level of rigor. The graphical representation of the statistical analysis of the pandemic in Greece is also presented in the following Figure 6 [29].

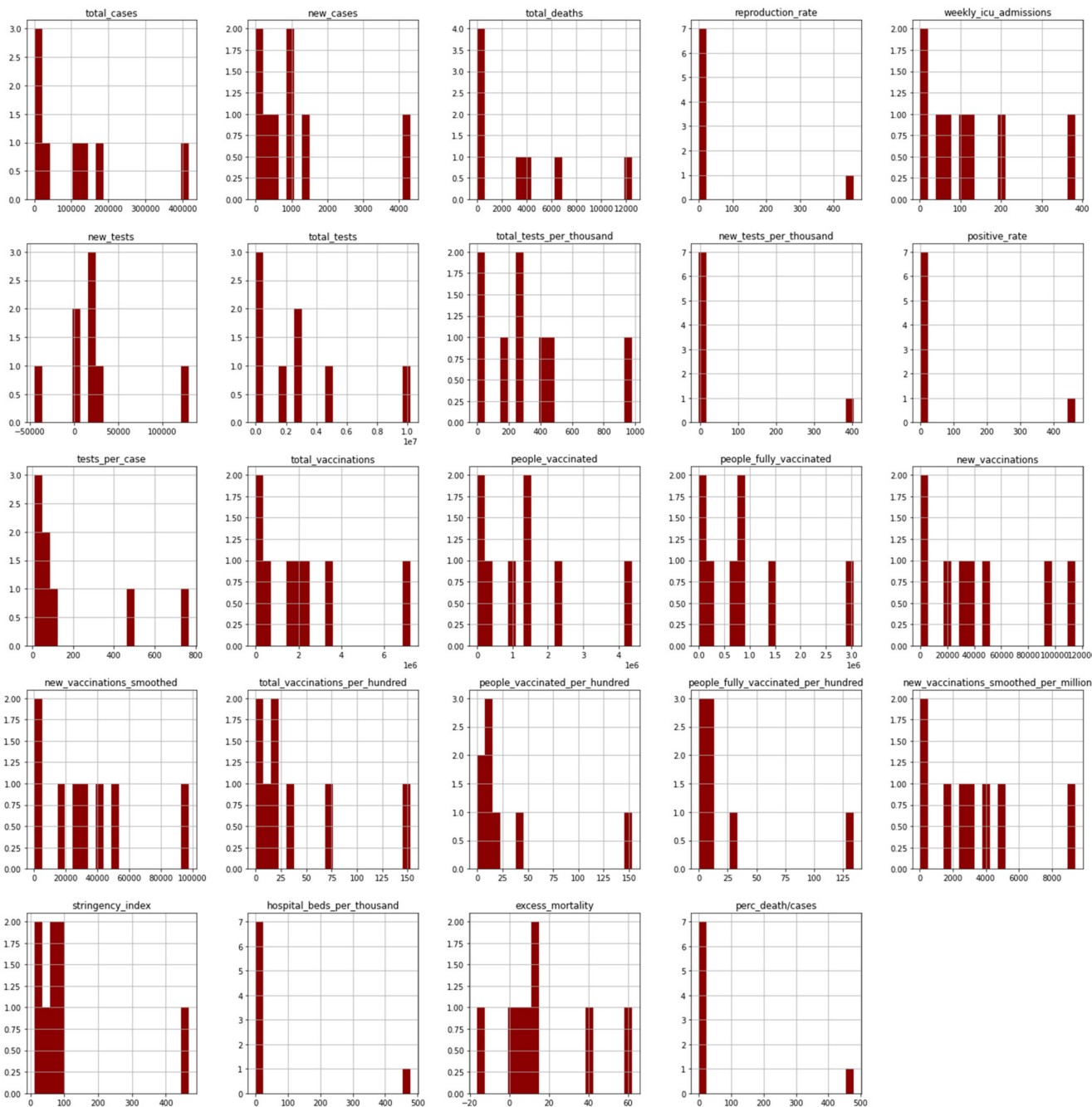

**Figure 6.** Pandemic statistical analysis of Greece.

The correlation between the above-examined variables of Tables 1–4 is presented in the following figure, and a table of the degree of Pearson correlation is defined in the Figure 7 [31]:

$$R = \frac{\sigma_{XY}}{\sigma_X \sigma_Y}. \tag{1}$$

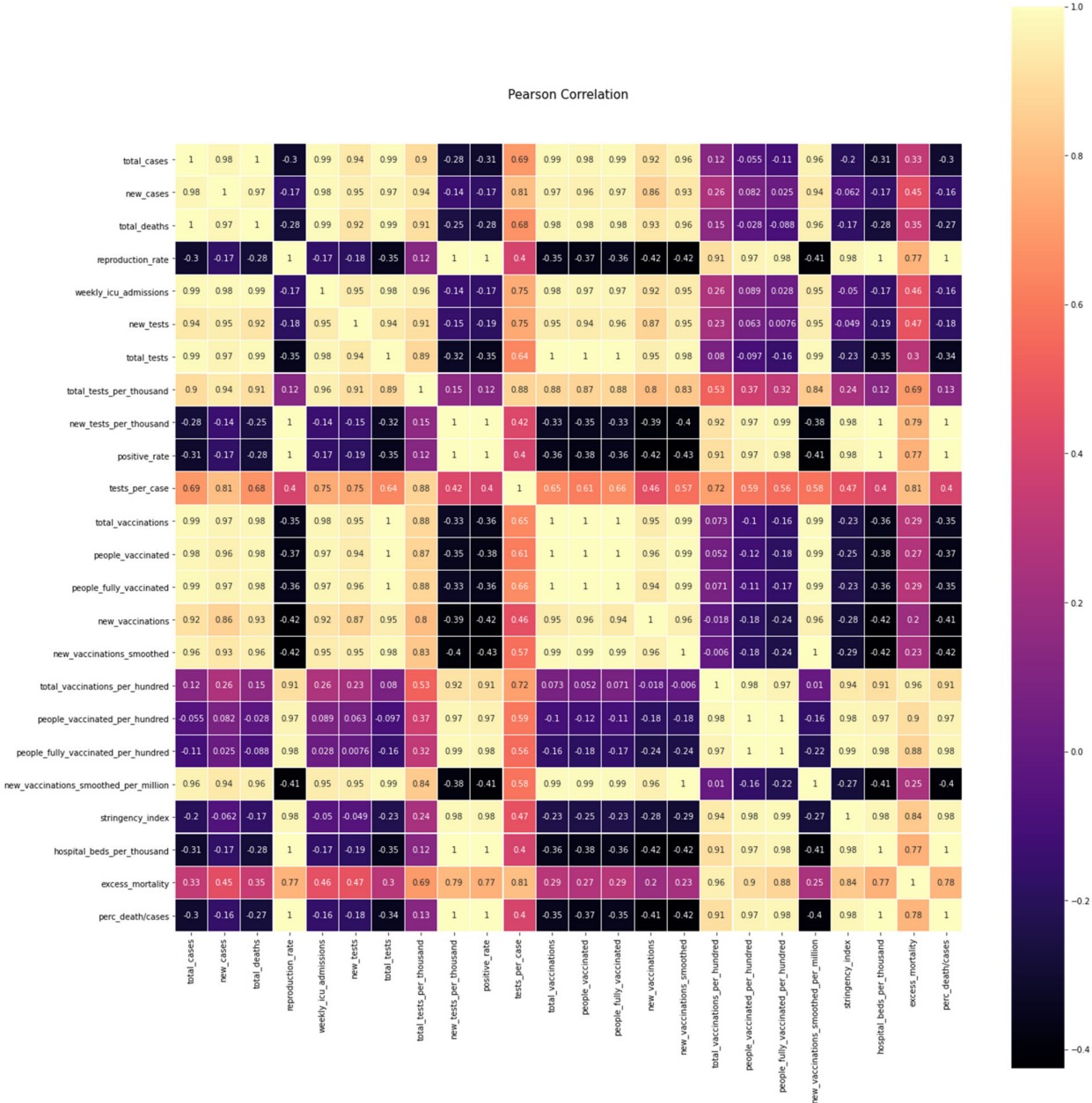

**Figure 7.** Pearson correlation matrix.

Essentially, the above table shows the degree of linear correlation of the variables $X$ and $Y$ with the dispersion of $\sigma_X^2$ and $\sigma_Y^2$ respectively and covariance $\sigma_{XY} = Cov(X, Y) = E(X, Y) - E(X)E(Y)$. The correlation coefficient $R$, similar to the covariance $\sigma_{XY}$, expresses the degree and the way the two variables are correlated, that is, how one random variable varies concerning the other. $\sigma_{XY}$ takes values that depend on the value range of $X$ and $Y$, while the coefficient R takes values in the interval $[-1, 1]$; where $R = 1$, there is a perfect positive correlation between $X$ and $Y$; if $R = 0$, there is no linear correlation between $X$ and $Y$; and if $R = -1$, there is a perfectly negative correlation between $X$ and $Y$. When $R = \pm 1$, the relation is causal and not probabilistic because knowing the value of one random variable, the exact value of the other variable is also known. When the correlation coefficient is close to $-1$ or 1, the linear correlation of the two variables is strong ($|R| > 0.9$), while when it is close to 0, the variables are practically unrelated [31].

*3.2. Near Real-Time Analytics*

From the moment the epidemic was identified as the result of the new coronavirus SARS-CoV-2, the main priorities of the scientific community were to collect appropriate data to be able to develop the most important parameters of descriptive epidemiology, which can model its evolution and spread disease, to make optimal decisions and ensure public health [19].

These data must be combined with epidemiological indicators related to the spread of COVID-19 disease, analyses for areas of interest that are directly related to the spread of the pandemic, as well as systems for recording and describing data such as tables, diagrams, etc. It should be emphasized that these mechanisms should not only be based on the logical results of the calculations performed but also on the time at which these results are available, because timing is a fundamental event in a real critical time system, such as the one under examination. Violation of time constraints implies the inability to make timely decisions and therefore implement incomplete measures that cannot work in a pandemic [6].

In this study, a thorough description of how the pandemic spread in Greece is presented [12], by presenting a data analysis system with machine learning methods, which was developed to capture in real time, taking into account the availability of data, statistics, correlations, charts, and comparative tables provided by official health agencies, plus any other relevant information related to the pandemic. The following Figures 8–13 show comparative diagrams with Greece's neighboring countries (Bulgaria, Albania, Turkey, and North Macedonia), aiming at assisting the decision-making system and the corresponding mechanisms of social distancing [7].

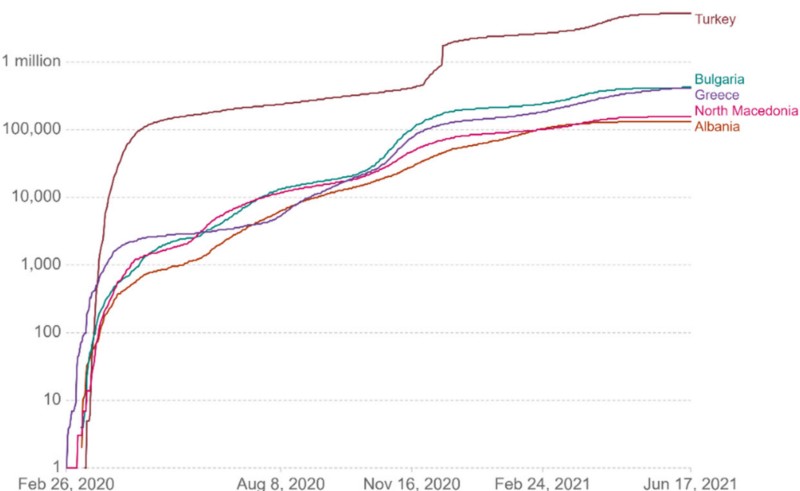

**Figure 8.** Cumulative confirmed cases per million.

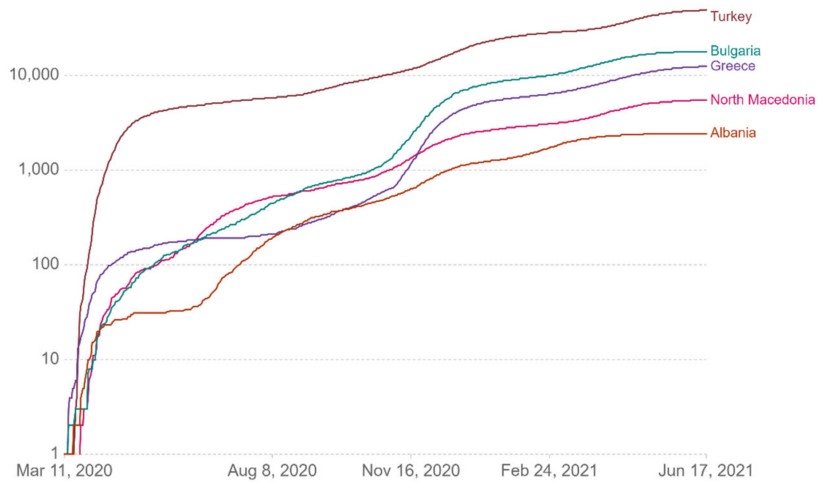

**Figure 9.** Cumulative confirmed deaths per million.

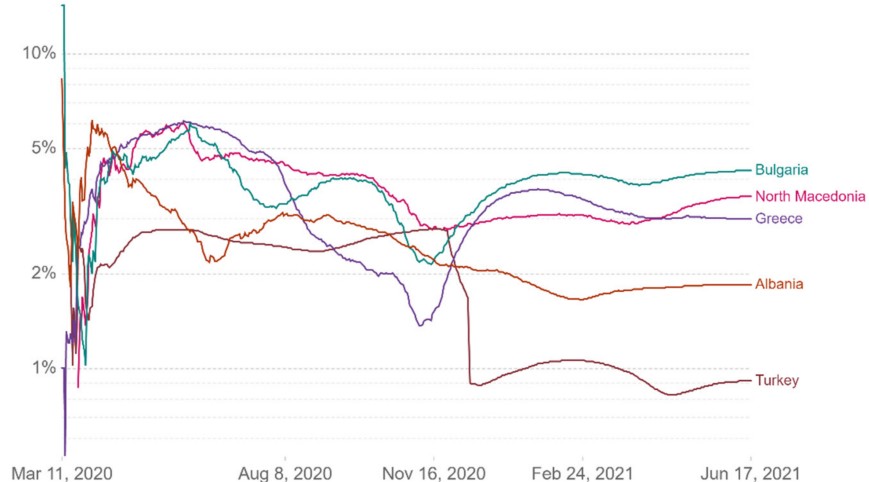

**Figure 10.** Case fatality rate.

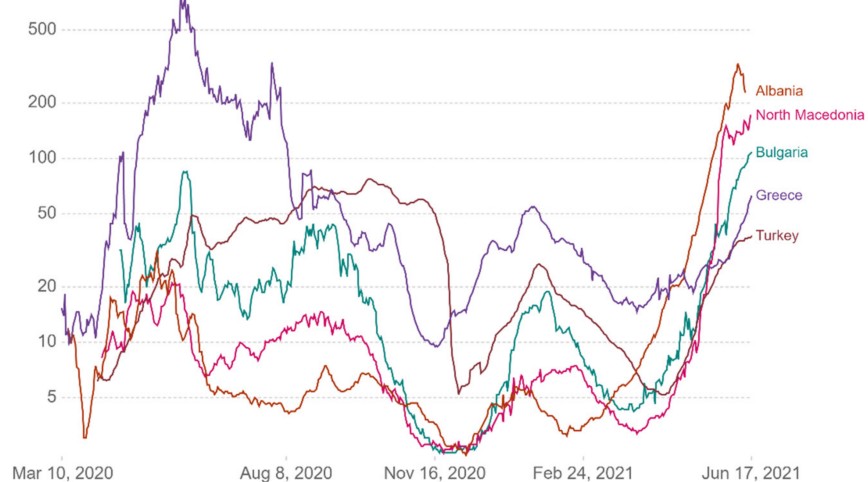

**Figure 11.** Cumulative tests per 1000 people.

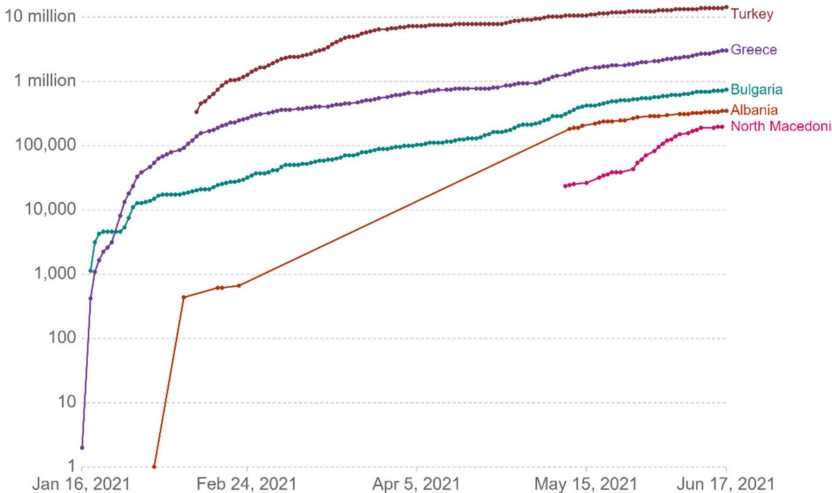

**Figure 12.** People fully vaccinated.

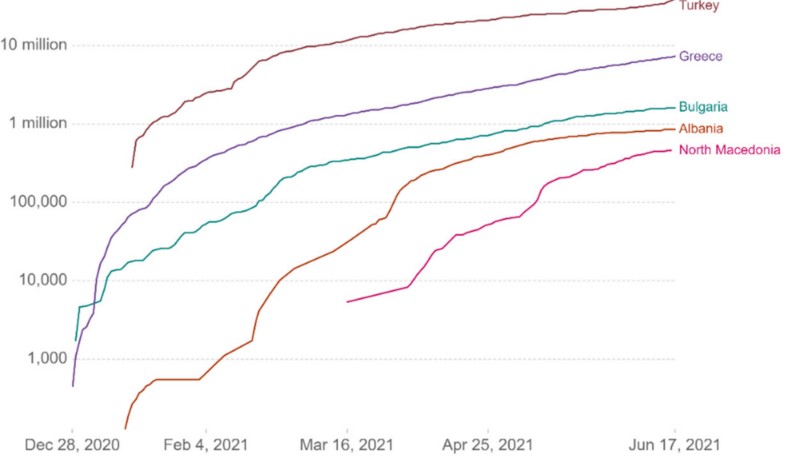

**Figure 13.** Vaccine doses administered.

In addition to a thorough analysis of the data provided, this system can calculate in real time the most important epidemiological indicators, which are presented below.

### 3.2.1. Basic Reproduction Number ($R_0$)

In epidemiology, $R_0$ can be thought of as the expected number of outbreaks at the beginning of an epidemic that results directly from an outbreak in a population where all individuals are susceptible to infection when there is no immunity in the population (natural or vaccinated) and no restrictive measures have begun to be implemented [27,28,32].

If, for example, $R_0 = 3$, each case can infect another three people on average, and these, in turn, another three each, and so on. As a result, the number of cases gradually increases, and there is an extensive dispersion. If $R_0 < 1$, then there is no risk of epidemic. This is because, in this case, one case can infect another person, and therefore, the transmission gradually declines. In general, the higher the value of $R_0$, the more difficult it is to control the epidemic. For simple models, the percentage of the population to be immunized to prevent the prolonged spread of the infectious disease must be greater than $1 - \frac{1}{R_0}$. On the other hand, the percentage of the population that remains prone to infection during the endemic equilibrium is $\frac{1}{R_0}$.

It is important to note that $R_0$ is not a biological constant for a pathogen, as it is also influenced by other factors, such as environmental conditions and the behavior of the infected population. In addition, $R_0$ does not in itself assess how quickly an infection is spreading in the population but should be considered in a broader research horizon. In

addition, the estimated values of $R_0$ depend on the model used and the values of other parameters, which suggests that the estimated values only make sense in the given space-time frame, and it is recommended not to use outdated values or to compare values based on different models [32].

### 3.2.2. Effective Reproduction Number ($R_t$)

When restrictive measures are implemented to reduce transmission, such as social distancing, the interest shifts from $R_0$ to $R_t$. This indicator expresses the number of people who can infect a case based on the restrictions imposed by the implementation of these restrictive measures [6,27,32].

This value may change over time as the gradual introduction of measures and the change in the behavior of the population (e.g., hand hygiene, contact restriction, etc.) make transmission increasingly difficult. The aim is to reduce it to $R_t < 1$, as this indicates that control of the epidemic has been achieved.

Monitoring the course of $R_t$ is extremely important, and its assessment should be updated at regular intervals based on the data collected from epidemiological surveillance (diagnosed cases per day) with the application of an appropriate methodology. In this way, the course of the epidemic and the effectiveness of the measures in real time can be approximated, since there is inevitably a delay from the moment a person becomes infected until he is diagnosed. Consequently, a possible increase in infections today could be reflected in the diagnosed cases of the coming days.

It is important to note that even if the epidemic has been reduced and the $R_t$ reduced to low levels, the stopping of the measures may lead to an increase of cases, which is a typical example we have seen in Greece. Therefore, in the phase of gradual phasing out of the measures, the monitoring of $R_t$ is very important as it will allow decisions to be taken for corrective actions if $R_t$ is approaching or exceeding the value of 1.

The first step in modeling the $R_t$ index is the input process of the recorded cases. A popular option for distributing these arrivals is to use the Poisson distribution, which is a distinct distribution function that expresses the probability of a given number of events occurring over a fixed period if these events occur by a known means rhythm and are independent of the time from the last case, as in the case under investigation. The Poisson distribution has the parameter $\lambda$ that indicates the average percentage of infections per day, which are independent of the last time of occurrence of the event, which is interpreted as the probability of occurrence of new cases every day and is given by the following function [26,28]:

$$P(k|\lambda) = \frac{\lambda^k e^{-\lambda}}{k!}. \tag{2}$$

Given the Poisson distribution, we can construct the probability distribution of new cases for a set of $\lambda_s$. The distribution of $\lambda$ on $k$ is called the probability function. The representation of the probability function by determining the number of new cases observed $k$ is calculated from the probability function in a range of values $\lambda$.

Under this relation, we can look for a new set $L(R_t | k_t)$, which parameterizes the relation between the Poisson distribution and the index $R_t$ and is expressed by the following relation [33,34]:

$$\lambda = k_{t-1} e^{\gamma(R_t - 1)} \tag{3}$$

where $\gamma$ is the inverse of the serial interval (about 4 days for COVID19) and $k_{t-1}$ is the number of new cases observed in time $t - 1$.

Since we know the exact number of cases per day, we can reformulate the probability function as Poisson, which is parameterized by specifying $k$ and changing $R_t$ and specifically as follows (Figure 14):

$$L(R_t|k) = \frac{\lambda^k e^{-\lambda}}{k!}. \tag{4}$$

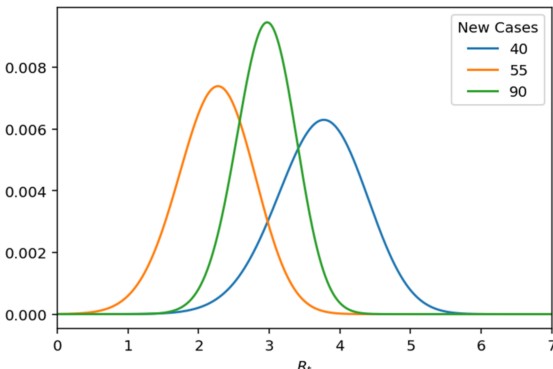

**Figure 14.** Likelihood of $R_t$ given $k$.

For each day, there is an independent conjecture about $R_t$. To combine the actual information from the previous days with the current day, Bayes' theorem is used to inform the hypotheses about the true value of $R_t$ based on the number of new cases reported daily. By this logic, Bayes' theorem is used as follows:

$$P(R_t|k_t) = \frac{P(R_t) \cdot L(k_t|R_t)}{P(k_t)}. \tag{5}$$

Using the probability of the previous period $P(Rt-1 \,|\, kt-1)$, the previous equation is written as follows:

$$P(Rt|kt) \propto P(Rt-1|kt-1) \cdot L(kt|Rt). \tag{6}$$

With iterative iterations up to $t = 0$, the relation becomes:

$$P(R_t|k_t) \propto P(R_0) \cdot \prod_{t=0}^{T} L(k_t|R_t). \tag{7}$$

With a uniform previous $P(R_0)$, this is reduced to:

$$P(R_t|k_t) \propto \prod_{t=0}^{T} L(k_t|R_t). \tag{8}$$

Taking the posterior probability, there is a significant change in the variance, as shown graphically in Figure 15 below.

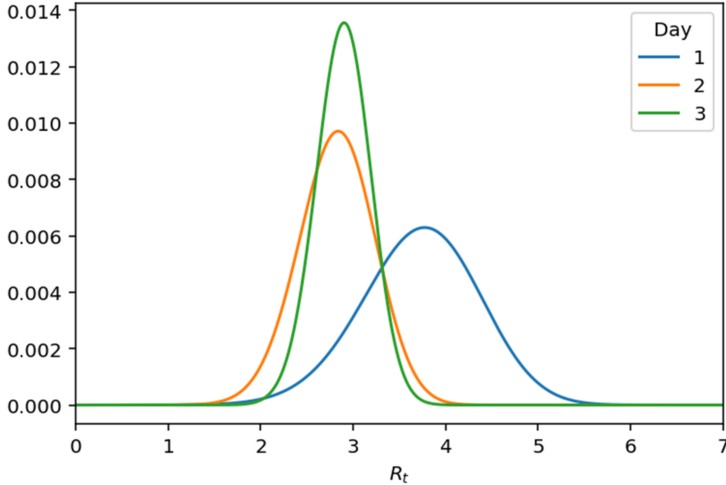

**Figure 15.** Posterior $P(R_t|k)$.

When estimating the quantity, it is very important to give a sense of the error surrounding the estimation. A popular way to do this is to use higher density intervals. This

calculation is done with the highest density interval (HDI) algorithm of posterior distributions. HDI can be used in the context of the uncertainty of classifying rear distributions as Credible Intervals (CI), where all points within this interval have a higher probability density than points outside the interval. With this parameterization, both the most probable values for the $R_t$ index and the HDI fluctuation over time can be plotted (Figure 16) [35].

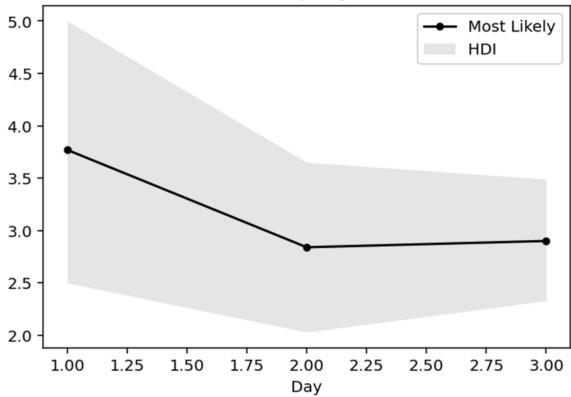

**Figure 16.** $R_t$ by day.

This is a very useful representation, as it shows how the components change every day. In essence, this view gives the most probable value of $R_t$, while expressing the certainty expressed over time, where the interval of the highest density decreases as the daily recorded cases increase. Below is captured each day (row) of the rear distribution that is designed simultaneously. The rear distributions start without much confidence (wide) and gradually become more confident (narrower) for the true value of $R_t$ (Figures 17–21).

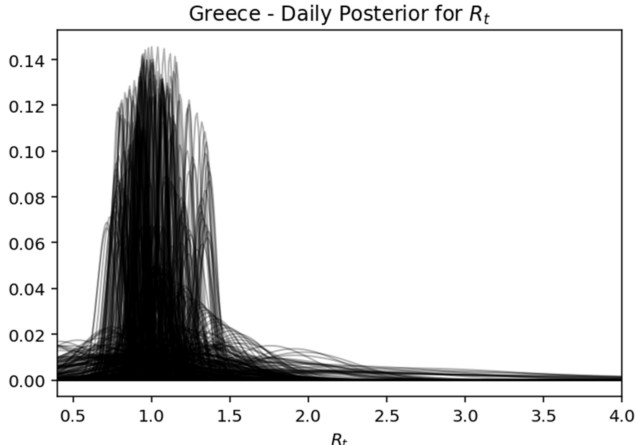

**Figure 17.** Greece—daily posterior for $R_t$.

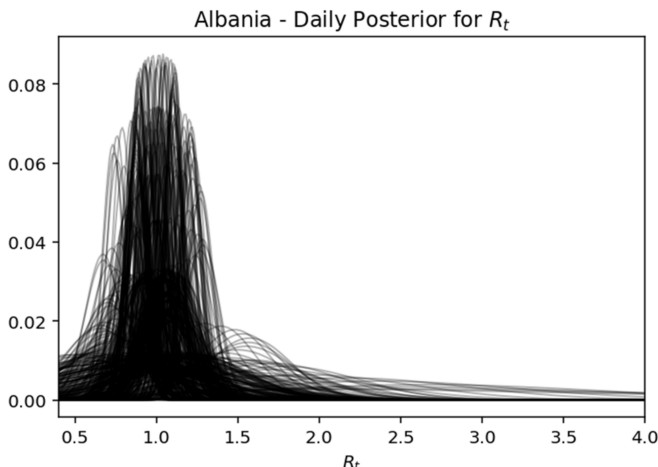

**Figure 18.** Albania—daily posterior for $R_t$.

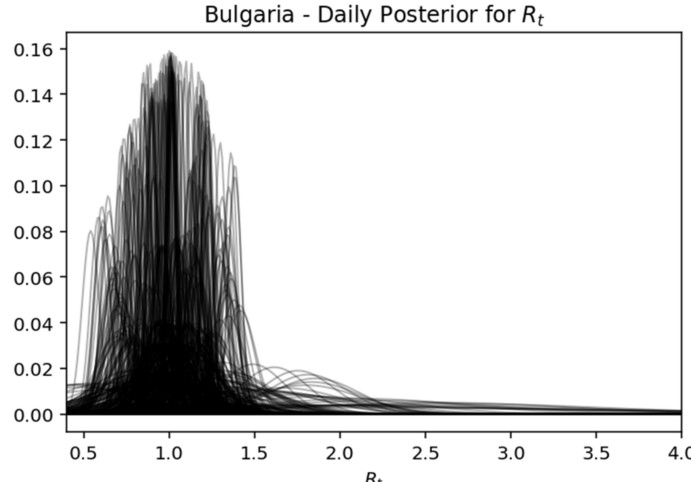

**Figure 19.** Bulgaria—daily posterior for $R_t$.

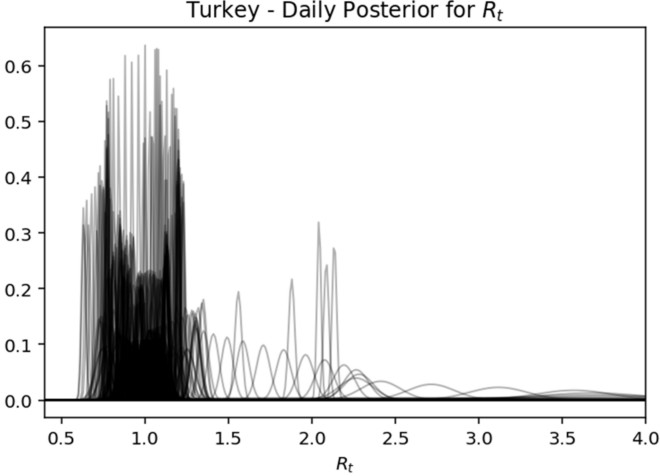

**Figure 20.** Turkey—daily posterior for $R_t$.

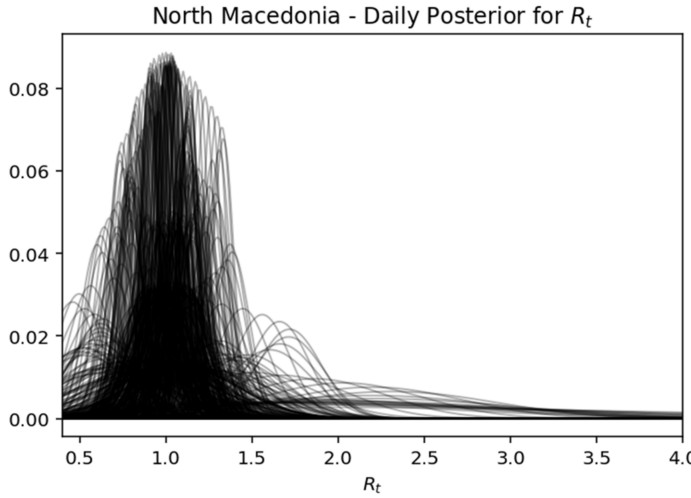

**Figure 21.** North Macedonia—daily posterior for $R_t$.

Since the results include uncertainty, it is desirable to show the most probable value of $R_t$ along with the higher density interval. In addition, taking into account the direct relationship that may exist in the spread of the virus with the opening of the borders and especially of the neighboring countries with land borders with Greece, this study includes similar studies for Albania, Bulgaria, Turkey, and North Macedonia, as shown in the Figure 22 below [11,28,32].

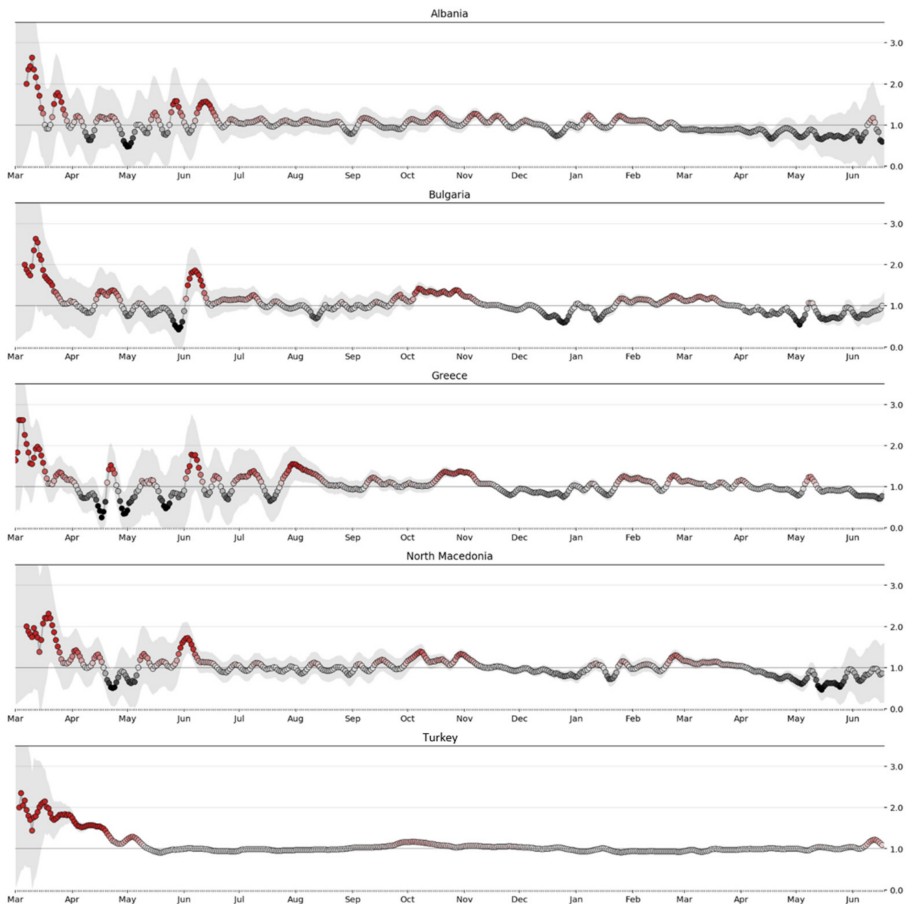

**Figure 22.** Real-time $R_t$ for Greece and surrounding countries.

Respectively in the following diagrams are presented detailed data on the variation of the $R_t$ for the examined countries and the probabilities related to the mentioned index (Figures 23 and 24) [6,11,33].

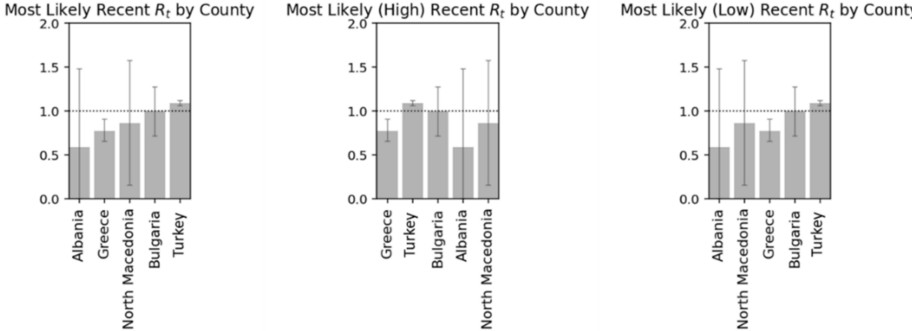

**Figure 23.** Most likely, high and low $R_t$ by country.

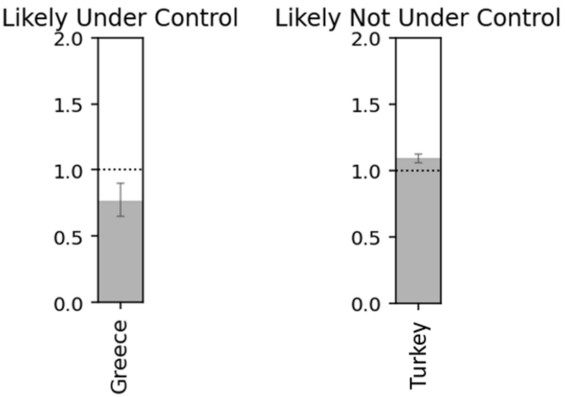

**Figure 24.** Countries $R_t$ with under and not under control.

The following Table 5 presents the index $R_t$ by country based on the statistical analysis for the most common values, as well as the respective Low and Max.

**Table 5.** Most likely value of $R_t$ along with its highest density interval.

| ID | Country | Most Likely | Low $R_t$ | Max $R_t$ |
|----|---------|-------------|-----------|-----------|
| 1 | Albania | 0.83 | 0.65 | 1.61 |
| 2 | Bulgaria | 0.89 | 0.61 | 1.16 |
| 3 | Greece | 0.74 | 0.62 | 0.84 |
| 4 | North Macedonia | 0.97 | 0.28 | 1.68 |
| 5 | Turkey | 0.98 | 0.68 | 1.94 |

### 3.2.3. Case Fatality Rate (CFR)

The CFR is the ratio of deaths from the virus to the total number of people diagnosed with the disease over a given period of time. It is essentially an assessment of the risk of death from the disease, and mortality is usually expressed as a percentage and is an indicator of the severity of the disease, while it is important to note that disease mortality is not stable. It varies between populations and varies over time, due to the interaction between the causative agent of the disease, the host, the environment, as well as the available treatment infrastructure and the quality of medical care resulting from the health system [27,32].

Reliable CFRs that can be used to assess deaths and evaluate any public health measures taken are calculated at the end of an epidemic, after resolving all cases related to

affected individuals who have either died or recovered. Figure 25 below shows the CFR index for Greece and its peripheral states.

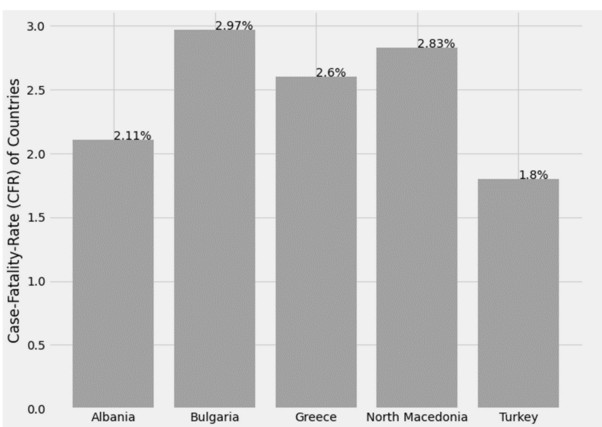

**Figure 25.** Case fatality rate by country.

3.2.4. Mortality Rate (MR)

Mortality or mortality rate is a measure of the number of deaths (either in general or due to a specific cause) in a given population, in terms of population size, per unit of time. As a rule, the unit of mortality is the number of deaths per 1000 people per year. The general form of the mortality calculation formula is $\frac{d}{p} \times 10^n$, where $d$ is the number of deaths from the cause being studied, $p$ is the size of the population from which the deaths came, and $10^n$ is a conversion factor that determines the size of the denominator. Specifically, the MR index is calculated as follows [6,32]:

$$Mortality_{Rate} = \frac{Confirmed_{Deaths}}{Confirmed_{Cases}}. \tag{9}$$

Figure 26 below shows the mortality rate index for the countries under study.

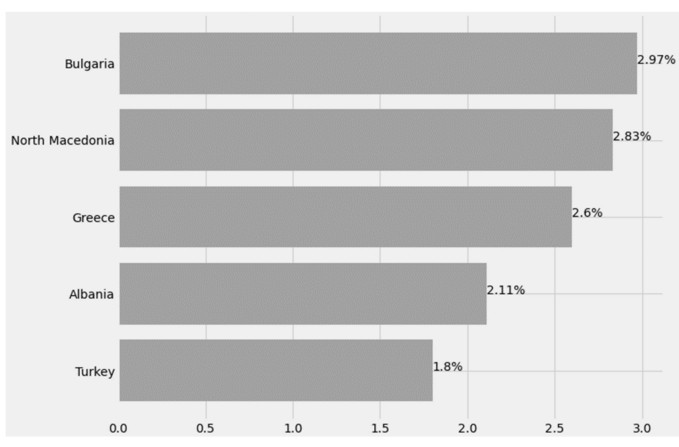

**Figure 26.** Mortality rate by country.

3.2.5. Recovery Rate (RR) or Discharge Rate (DR)

In its simplest form, the RR is calculated by dividing the number of recoveries by the number of confirmed cases. Specifically, the RR index is calculated as follows [27,28,32,33]:

$$Recovery_{Rate} = \frac{Recovered_{Cases}}{Confirmed_{Cases}}. \tag{10}$$

Figure 27 below shows the RR index for the countries under study.

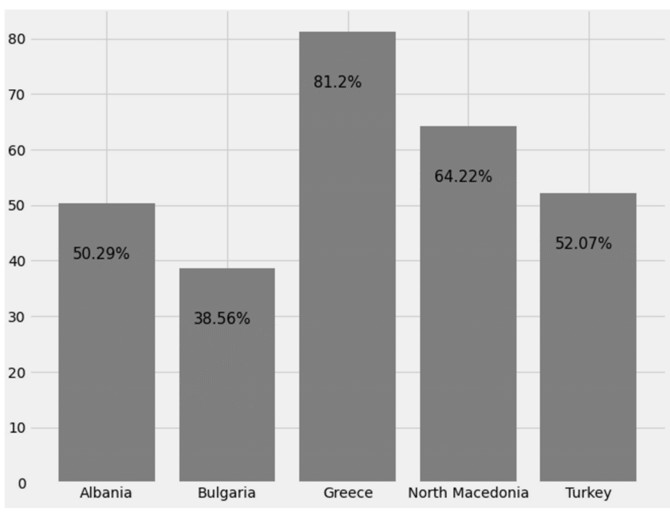

**Figure 27.** Recovery rates by country.

### 3.2.6. Infection Rate (IR)

IR is the apparent rate of infection, which is an estimation of the rate of disease progression, based on proportional measures of the extent of infection at different times.

Initially, a proportional measure of the extent of the infection is chosen as a measure of the extent of the disease. Then, measurements of the extent of the disease are taken over time, based on an appropriate mathematical model. The model is based on the assumption that the progression of the infection is limited by the amount of the population remaining to be infected, in which case the extent of the infection is limited, and otherwise, it would show exponential growth. A model of its calculation can be calculated in detail using the following formula [26,28,32]:

$$IR = \frac{1}{t_2 - t_1} \log_e \left[ \frac{x_2(1 - x_1)}{x_1(1 - x_2)} \right] \qquad (11)$$

where $t_1$ is the time of the first measurement, $t_2$ is the time of the second measurement, $x_1$ is the proportion of infection measured at time $t_1$, and $x_2$ is the proportion of infection measured at time $t_2$. The values for the maximum infection rate of the study countries are presented in the Table 6 below [6,28,32,33].

**Table 6.** Maximum infection rates.

| ID | Country | Max IR |
|----|---------|--------|
| 1 | Albania | 1239.0 |
| 2 | Bulgaria | 4828.0 |
| 3 | Greece | 3316.0 |
| 4 | North Macedonia | 1402.0 |
| 5 | Turkey | 82,325.0 |

### 3.2.7. Prevalence

This is the proportion of a specific population that is found to be affected by the epidemic and essentially expresses the actual number of patients in the population. It comes from comparing the number of people found to have the disease with the total number of people studied and is usually expressed as a fraction, percentage, or the number of cases per 10,000 or 100,000 people. Point prevalence is the proportion of a population that has the disease at a given time, while period prevalence is the proportion of a population that has the disease at any given time in a given period (e.g., twelve-month prevalence).

Lifetime prevalence is the proportion of a population that at some point in its life (up to the time of assessment) has been affected by the disease (Figure 28) [26,27,32].

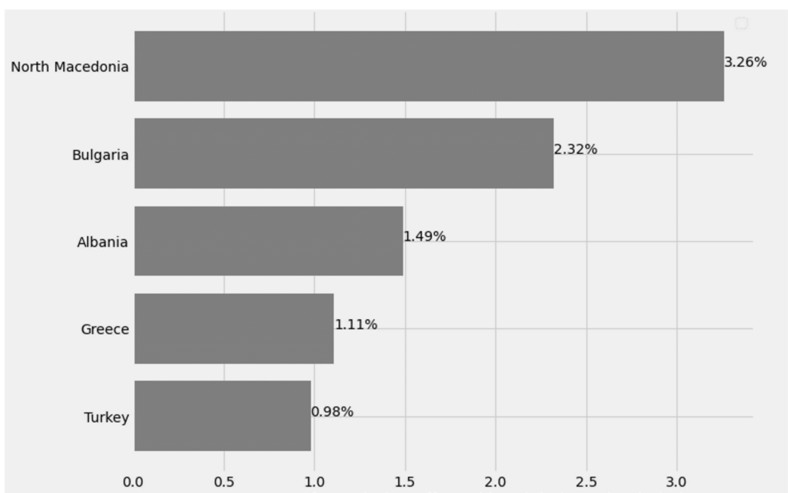

**Figure 28.** Percentage of population affected by global pandemic.

## 4. Prediction Model

Making a decision is a complex process, which must take into account many different factors. As part of an ideal process, information should be gathered on all the possible factors involved, the weight and influence of each factor should be understood, an exhaustive list and meticulous study of all possible solutions should be made, and the benefits and costs for each of them should be assessed. Such an ideal process yields the optimal solution [6,14].

The ability to accurately predict the course of the pandemic is an extremely important but difficult task. Due to the limited knowledge of the new COVID-19 disease, the high uncertainty, and the complex socio-political factors that affect the spread of the new virus, the constant information and any scientifically substantiated methodology of analysis or prediction of the phenomenon is an important legacy.

Focusing on the specifics of the spread of the disease, both epidemiologically and in terms of implementation of preventive and repressive measures, this paper conducts an exploratory study, which is based on the analysis of time-series data related to COVID-19 disease and the prediction of the future development of the pandemic for Greece but also for the border countries.

To accurately approach the problem, the goal is to find the mathematical relationship that can model the data on the spread of the disease and how the cases increase over time. Facebook's Prophet, an innovative and highly reliable time series prediction model, was used as the forecasting methodology.

Prophet is based on the general methodology of Generalized Additive Models (GAM) [36–38], which is a modeling method that uses non-parametric techniques offering significant advantages over conventional regression methods. That is, it offers an opportunity to overcome the statistical problems associated with the normality and linearity assumptions that are necessary for linear regression.

The name Additive refers to the multivariate hypothesis of the underlying model, according to which the predictors have a cumulative structure. Such models are interesting if they fit the data because they are easier to interpret. In general, a cumulative regression model uses cumulative adaptive methods for modeling. Thus, the researcher is not required to look for the correct transformation of each variable.

More specifically, the estimation of the dependent variable $Y$ in this case for a single independent variable can be given by the following equation [37,38]:

$$Y = s(X) + error \tag{12}$$

where $s(X)$ is an unspecified smoothing function, while error is the error that usually has zero mean value and constant dispersion. For example, the smoothing function can be determined by the current mean or by the current median or by the local least squares method, the Kernel method, the Loess method, or the spline method. The term current means the serial calculation of a statistic applied to overlapping intervals of values of the independent variable, such as the running mean. In GAM modeling, the classical linear hypothesis is extended to include any probability distribution (Poisson, Gamma, Gaussian, Binomial, and Inverse Gaussian) error by the exponent group.

Similar to a GAM, with time as a regressor, Prophet can adapt to many linear and non-linear functions of time as components, wherein its simplest form, three basic elements are used: trend, seasonality, and holidays, which are combined in the following equation [39,40]:

$$y(t) = g(t) + s(t) + h(t) + e(t) \tag{13}$$

where:

1. $g(t)$, trend models non-periodic changes (i.e., growth over time)
2. $s(t)$, seasonality presents periodic changes (i.e., weekly, monthly, yearly)
3. $h(t)$, ties in effects of holidays (on potentially irregular schedules $\geq 1$ day(s))
4. $e(t)$, covers idiosyncratic changes not accommodated by the model

In general, the whole equation can be written as follows:

$$y(t) = piecewise\_trend(t) + seasonality(t) + holiday\_effects(t) + noise(t). \tag{14}$$

In a more thorough analysis, the test variables can be structured as follows:

1. Trend. The process includes two possible trend models for $g(t)$, namely a Saturating Growth Model and a piecewise linear model as follows [39,40]:

a. Saturating Growth Model. If the data suggests promise of saturation:

$$g(t) = \frac{C}{1 + exp(-k(t-m))} \tag{15}$$

where $C$ is the carrying capacity, $k$ is the growth rate, and $m$ is an offset parameter.

It is possible to incorporate trend changes in the model, explicitly specifying the change points where the growth rate change is allowed. Assuming that there are $S$ change points during periodic $s_j$, $j = 1, \ldots, S$, then Prophet defines a vector of $\delta_j$ rate change settings in time $s_j$, with $\delta \in R^S$. So, at any time $t$, the rhythm $k$ can be formulated as $k + \sum_{j:t>s_j} \delta_j$. If in this relation, the vector $\alpha(t) \in \{0,1\}^S$ is also determined, so that:

$$a_j(t) = \begin{cases} 1, \ if \ t \geq S_j \\ 0, \ otherwise \end{cases}, \tag{16}$$

then, the rhythm at the moment $t$ is $k + a(t)\delta$. When the rate $k$ is adjusted, the offset parameter $m$ must also be adjusted to connect the endpoints of the sections. The correct setting at the change point $j$ is easily calculated as:

$$y_j = \left(S_j - m - \sum_{i<j} y_t\right)\left(1 - \frac{k + \sum_{i<j} \delta_t}{k + \sum_{i \leq j} \delta_t}\right). \tag{17}$$

The final function is completed as follows:

$$g(t) = \frac{C(t)}{1 + exp(-(k + a(t)\delta)(t - (m + a(t)y)))}. \tag{18}$$

b. Linear Trend with Changepoints. This is a Piecewise Linear Model with a constant growth rate, which is calculated as follows:

$$g(t) = (k + a(t)\delta)t + (m + a(t)y) \tag{19}$$

where $k$ is the growth rate, $\delta$ has the rate adjustments, $m$ is the offset parameter, and to make the function continuous, $y_j$ is set to $-S_j\delta_j$.

c. Automatic Changepoint Selection. To identify changepoints, it is recommended to identify a large number of changepoints as follows:

$$\delta_j \sim Laplace(0, \tau) \tag{20}$$

where $\tau$ directly controls the flexibility of the model in altering its rate. It should be noted that a sparse previous adjustment $\delta$ has no effect on the primary growth rate $k$, so it progresses to 0, and the adjustment reduces the typical (no piecewise) logistic or linear growth.

d. Trend Forecast Uncertainty.

When the model deviates beyond the background to make a prediction, the trend $g(t)$ will have a steady pace. Uncertainty in the forecast trend is assessed by extending the production model forward where there are $S$ change points over a history of points $T$, each of which has a change of pace $\delta_j \sim Laplace(0, \tau)$ derived from the data, which is achieved by estimating the maximum probability of the rate scale parameter as follows:

$$\lambda = \frac{1}{S}\sum_{j=1}^{S}|\delta_j|. \tag{21}$$

Future sample change points are randomized in such a way that the mean frequency of change points matches the corresponding historical points as follows:

$$\forall_j > T, \begin{cases} \delta_j = 0 \ w.p. \ \frac{T-S}{T} \\ \delta_j \sim Laplace(0, \lambda) \ w.p. \ \frac{S}{T} \end{cases}. \tag{22}$$

2. Seasonality. The seasonal variable $s(t)$ provides adaptability to the model allowing periodic changes based on daily, weekly, and annual seasonality. Prophet relies on the Fourier series to provide a flexible model of periodic modeling, where approximately arbitrarily smooth seasonal snapshots are associated with a typical Fourier series:

$$s(t) = \sum_{n=1}^{N}\left(a_n \cos\left(\frac{2\pi nt}{P}\right) + b_n \sin\left(\frac{2\pi nt}{P}\right)\right). \tag{23}$$

3. Holidays and Events. The item $h(t)$ reflects predictable events of the year, including those on irregular schedules, which, however, create serious bias in the model. Assuming that the holiday effects are independent, seasonality is calculated by the model creating a regression matrix:

$$Z(t) = [1(t \in D_1), \dots, 1(t \in D_L)]$$
$$h(t) = Z(t)k. \tag{24}$$

## 5. Data and Results

The data used to mathematically model and predict disease spread are freely available for use at the COVID-19 data repository by the Center for Systems Science and Engineering at Johns Hopkins University [29], and they include the daily measurements during the period from 26 February 2020 to 31 May 2021 of the total recorded cases.

With an initial approach to measurements related to the spread of COVID-19 disease, we find that this is a dataset that is collected over time and expresses the evolution of values over equal successive periods (daily measurements). In particular, it is a continuous-time

series, where the price trend is initially upward, while there are intervals that show signs of stability.

Respectively, no fluctuations of the values that vary with time were found, as the time series does not show periodic fluctuations or changes that occur due to exogenous factors during specific periods. Although the test sample is not large enough, the above two tests confirm that the time variation of COVID-19 disease is recorded with data that are part of a static time series.

With a more thorough analysis, we look on the one hand for those characteristics that focus on estimating the system that produces the time series and on the other hand at finding the corresponding characteristics that contribute to understanding the historical behavior of the disease, thus allowing the prediction of its future prices.

In attempting to predict the spread of the disease in Greece, the Prophet algorithm was applied [39]. Specifically, considering all the pairs $(x_1, f(x_1)), \ldots, (x_n, f(x_n))$ of arithmetic figures of the spread of the disease in Greece, the proposed forecasting system aims to calculate an optimal approach to the spread of the pandemic $x_0 \in R$ with $x_0 \neq x_i$, $i \in \{1, \ldots, n\}$ so that the estimated $\hat{f}(x_0)$ is as close as possible to the real $f(x_0)$. The main objective of the process is to calculate the value $\hat{f}(x_0)$ for $x_0 \neq x_i$, $i \in \{1, \ldots, n\}$, for generalization purposes, i.e., the implementation of a realistic model that will not be completely guided by the historical data, which are its reference point.

Given the fact that the time series under consideration has a constant rate of change, the Prophet algorithm was used for the daily forecast from 1 June to 1 September 2021, using as training data the daily cases from 26 February 2020 to 28 February 2020 (369 days) and as a set of testing and confirmation of the model the period from 1 March 2021 to 31 May 2021 (92 days).

The following metrics were used to confirm the result:

1. Coefficient of Determination—$R^2$ [31]. To express the correlation of two random variables, $R^2$ is used, which is expressed as a percentage (%). It gives the percentage of variability of $Y$ values calculated from $X$ and vice versa and is a useful way to accurately determine the correlation of two random variables. $R^2$ is defined as follows:

$$R^2 = 1 - \frac{\sum_{i=1}^{n} (Y_i - \hat{Y}_i)^2}{\sum_{i=1}^{n} (Y_i - \overline{Y}_i)^2} \tag{25}$$

where $Y_i$ represents the observed values of the dependent variable, $\hat{Y}_i$ represents the estimated values of the dependent variable, $\overline{Y}$ represents the arithmetic mean of the observed values, and n represents the number of observations. $R^2$ expresses the percentage of variability of the dependent variable explained by the existence of independent variables in the model and takes values in the interval [0, 1], with optimal performance when its value approaches the unit, which is interpreted that then, the regression model adapts optimally to the data.

2. Root Mean Squared Error—RMSE [31]. The RMSE is directly related to the Standard Error of the Regression (SER) and calculates the average error of the predicted values about the actual values. It is calculated based on the following formula:

$$RMSE = \sqrt{\frac{1}{n} \sum_{j=1}^{n} \left( P_{(ij)} - T_j \right)^2} \tag{26}$$

where $P_{(ij)}$ is the value predicted by the program $i$ for a simple hypothesis $j$ and $T_j$ is the target value for the simple hypothesis $j$. The success of a regression model requires extremely small values for the root of the mean square error, while the best case, which implies an absolute correlation between actual and predicted values and therefore the absolute success of the model, is achieved when $P_{(ij)} - T_j = 0$.

Mean Absolute Error—MAE. The MAE is the measure of quantification of the error between the estimate or forecast to the observed values. It is calculated by the formula:

$$MAE = \frac{1}{n}\sum_{i=1}^{n}|f_i - y_i| = \frac{1}{n}\sum_{i=1}^{n}|e_i| \tag{27}$$

where $f_i$ is the estimated value and $y_i$ is the true. The average of the absolute value of the quotient of these values is defined as the absolute error of their relation $|e_i| = |f_i - y_i|$.

3. Mean Absolute Percentage Error—MAPE [31]. The average percentage absolute difference provides an objective measure of the forecast error as a percentage of demand (e.g., the forecast error is on average 10% of actual demand), without depending on the order of magnitude of demand. It is calculated by the formula:

$$MAPE = 100\sum_{t=1}^{T}\frac{\left[\frac{|D_t - F_t|}{D_t}\right]}{T}. \tag{28}$$

The results are presented in the Table 7 below.

**Table 7.** Performance metrics of the Prophet method.

| Prophet | $R^2$ | RMSE | MAE | MAPE |
|---|---|---|---|---|
| | 99,998 | 2.259 | 1.357 | 0.179 |

Diagrams of the methodology that show how the algorithm works and respectively how the problem is modeled are presented in the following Figures 29–36.

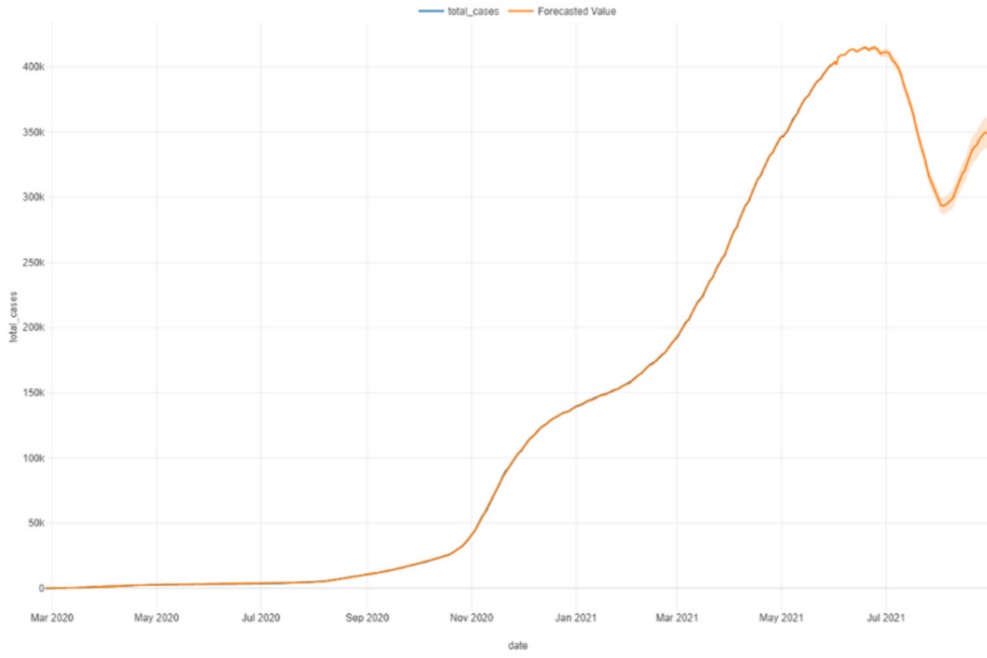

**Figure 29.** Prophet forecast.

The diagram of the process including the trend changes is presented in the following image.

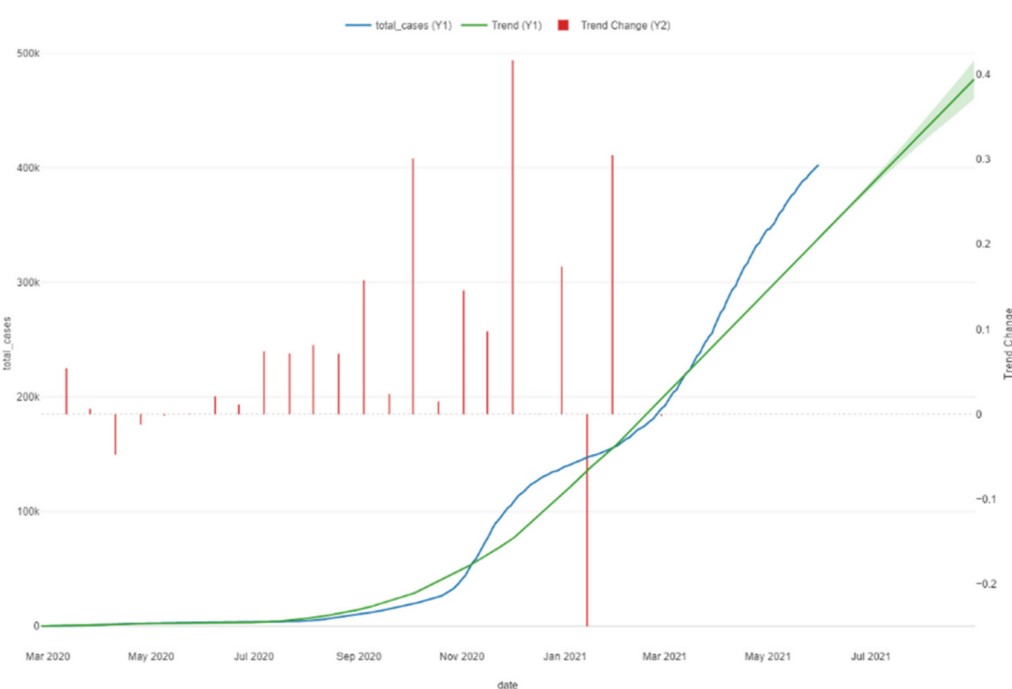

**Figure 30.** Prophet forecast with trend changes.

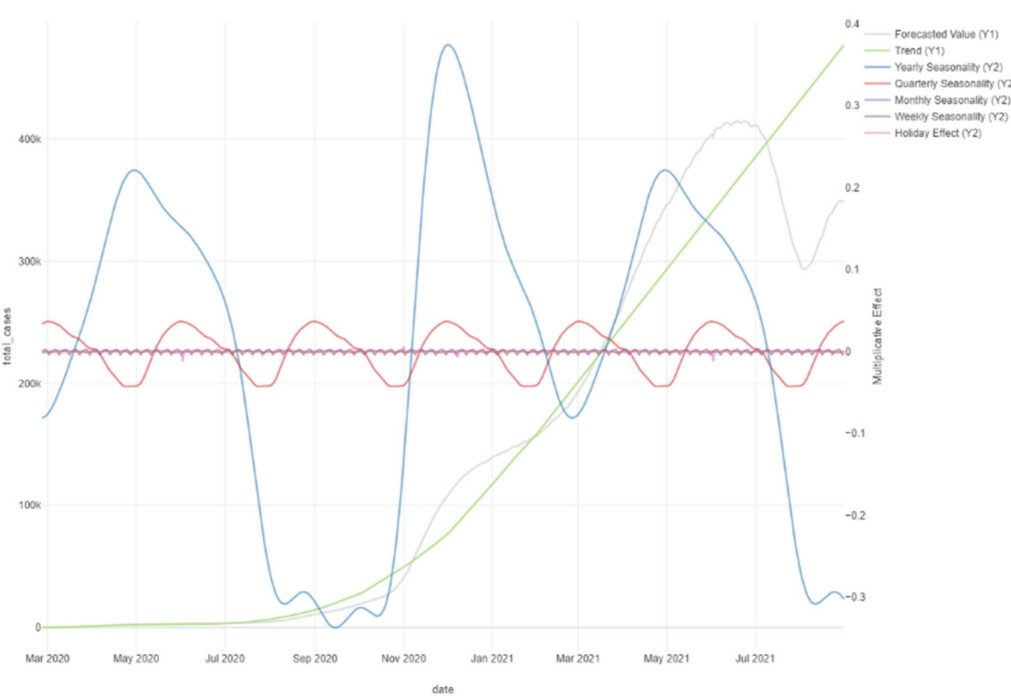

**Figure 31.** Prophet forecast effects.

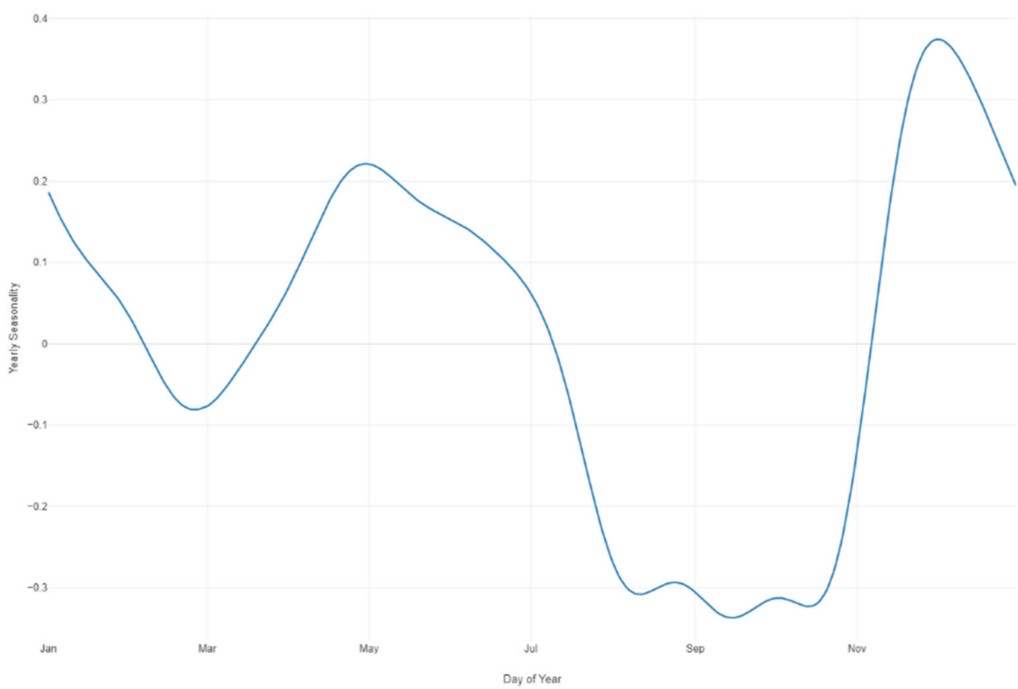

**Figure 32.** Prophet forecast yearly seasonality.

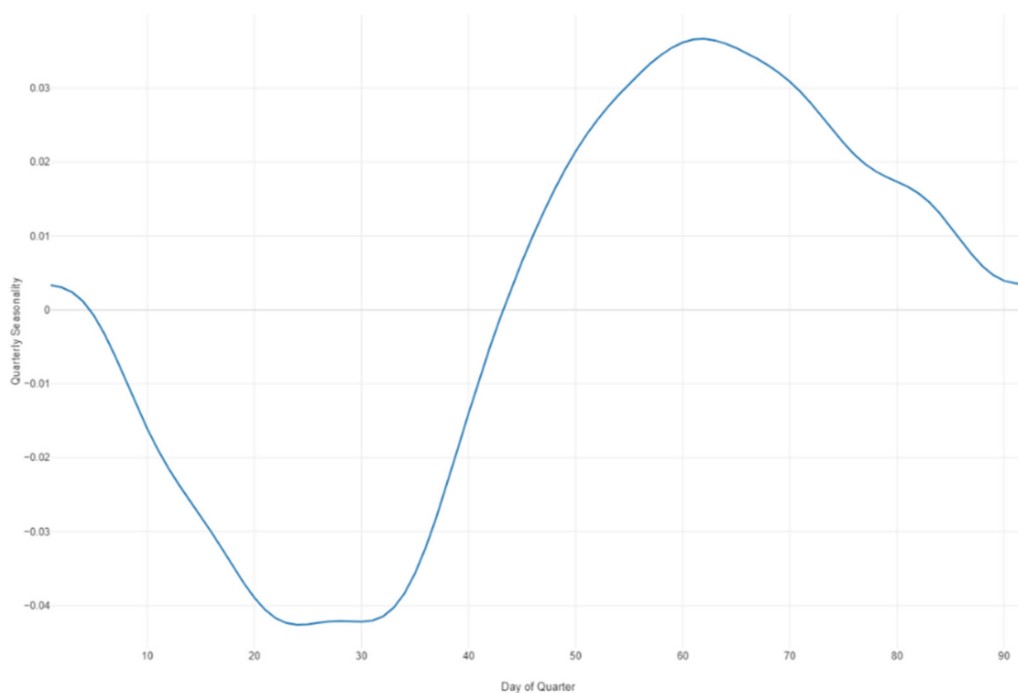

**Figure 33.** Prophet forecast quarterly seasonality.

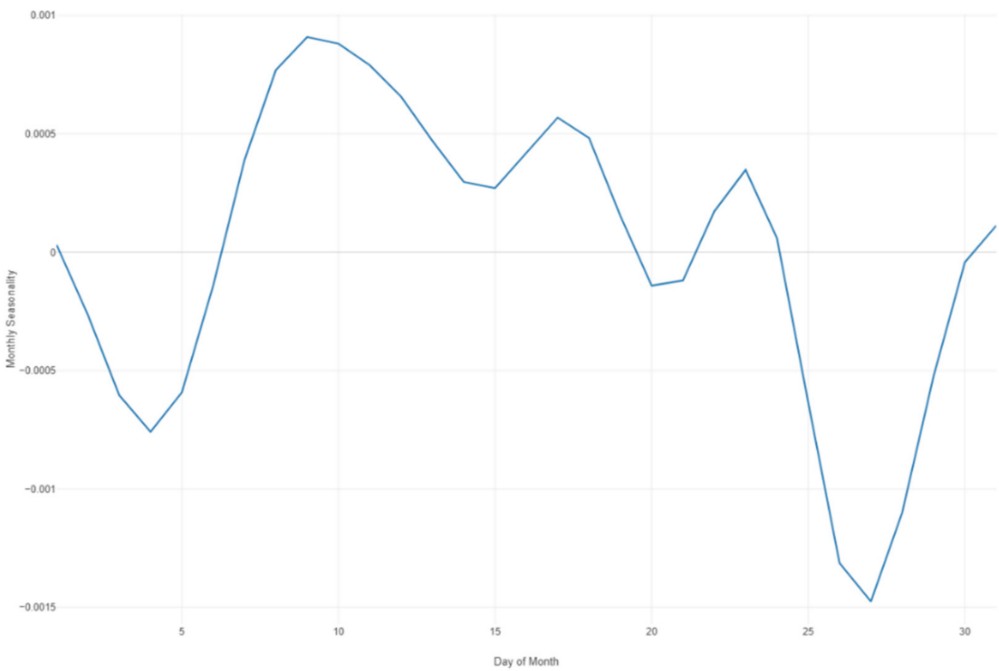

**Figure 34.** Prophet forecast monthly seasonality.

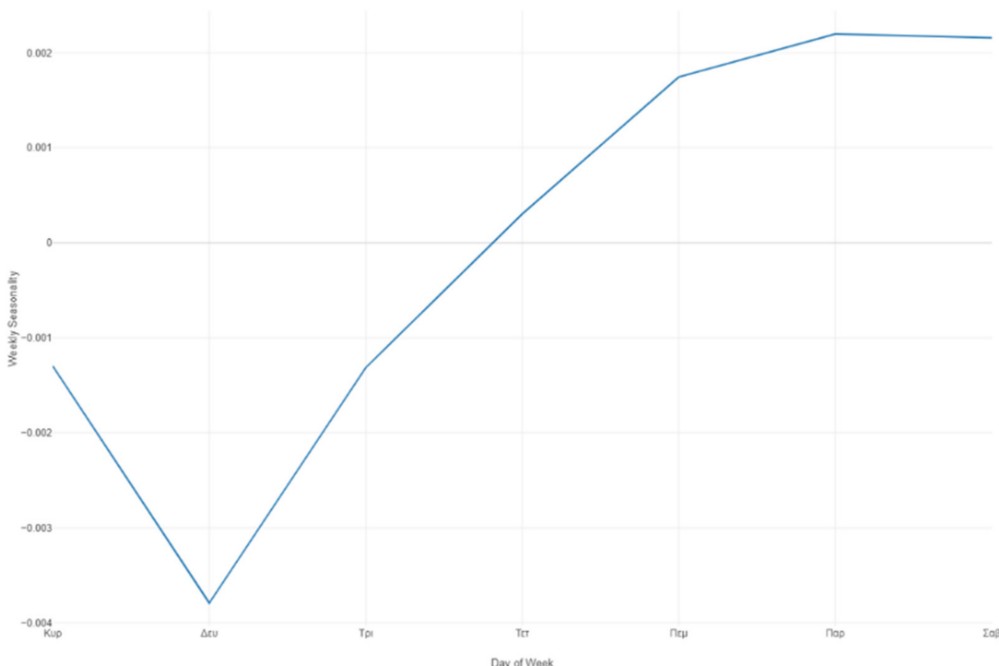

**Figure 35.** Prophet forecast weekly seasonality.

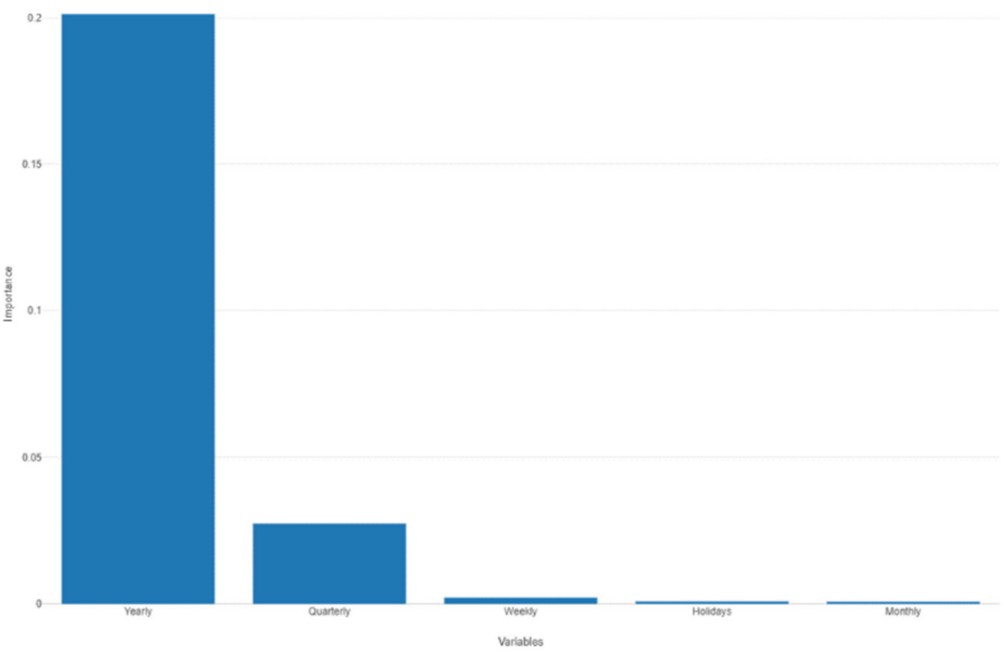

**Figure 36.** Prophet forecast importance seasonality.

Finally, the total table of detailed forecasts of the methodology from 1 June 2021 to 1 September 2021 is presented in the Table 8 below.

**Table 8.** Prophet forecast values.

| Date | Forecasted | | | Trend | | | |
|---|---|---|---|---|---|---|---|
| | Value | High | Low | Yearly | Quarterly | Monthly | Weekly |
| 1/6/2021 | 404,185 | 405,914 | 405,495 | 0.15301 | 0.03672 | 0.00013 | −0.00131 |
| 2/6/2021 | 405,563 | 406,259 | 406,069 | 0.15132 | 0.03662 | −0.00017 | 0.0003 |
| 3/6/2021 | 406,241 | 408,937 | 407,573 | 0.14958 | 0.03626 | −0.00057 | 0.00174 |
| 4/6/2021 | 407,386 | 409,059 | 407,990 | 0.14776 | 0.03572 | −0.00083 | 0.0022 |
| 5/6/2021 | 408,414 | 410,570 | 408,029 | 0.14586 | 0.03505 | −0.00073 | 0.00216 |
| 6/6/2021 | 409,442 | 411,155 | 409,148 | 0.14387 | 0.0343 | −0.00029 | −0.0013 |
| 7/6/2021 | 410,470 | 412,972 | 410,067 | 0.14178 | 0.03347 | 0.00031 | −0.00379 |
| 8/6/2021 | 411,498 | 413,968 | 411,286 | 0.13959 | 0.03254 | 0.00076 | −0.00131 |
| 9/6/2021 | 412,526 | 415,169 | 412,305 | 0.13729 | 0.03145 | 0.00093 | 0.0003 |
| 10/6/2021 | 413,554 | 416,370 | 413,124 | 0.1349 | 0.03016 | 0.0009 | 0.00174 |
| 11/6/2021 | 414,582 | 417,570 | 413,943 | 0.13241 | 0.02866 | 0.00082 | 0.0022 |
| 12/6/2021 | 415,610 | 418,771 | 414,762 | 0.12982 | 0.02697 | 0.00071 | 0.00216 |
| 13/6/2021 | 416,638 | 419,972 | 415,581 | 0.12714 | 0.02517 | 0.00052 | −0.0013 |
| 14/6/2021 | 417,666 | 421,173 | 416,400 | 0.12438 | 0.02337 | 0.00029 | −0.00379 |
| 15/6/2021 | 418,694 | 422,374 | 417,219 | 0.12155 | 0.0217 | 0.00021 | −0.00131 |
| 16/6/2021 | 419,722 | 423,575 | 418,038 | 0.11864 | 0.02028 | 0.00037 | 0.0003 |
| 17/6/2021 | 420,750 | 424,776 | 418,857 | 0.11566 | 0.01915 | 0.00061 | 0.00174 |
| 18/6/2021 | 421,778 | 425,977 | 419,676 | 0.11262 | 0.0183 | 0.0006 | 0.0022 |
| 19/6/2021 | 422,806 | 427,178 | 420,495 | 0.10951 | 0.01763 | 0.00024 | 0.00216 |
| 20/6/2021 | 423,834 | 428,379 | 421,314 | 0.10634 | 0.017 | −0.00018 | −0.0013 |
| 21/6/2021 | 424,862 | 429,580 | 422,133 | 0.10309 | 0.01624 | −0.00024 | −0.00379 |
| 22/6/2021 | 425,890 | 430,781 | 422,952 | 0.09975 | 0.01521 | 0.00012 | −0.00131 |
| 23/6/2021 | 426,918 | 431,982 | 423,771 | 0.09632 | 0.01383 | 0.00046 | 0.0003 |
| 24/6/2021 | 427,946 | 433,183 | 424,591 | 0.09277 | 0.01213 | 0.00029 | 0.00174 |

**Table 8.** *Cont.*

| Date | Forecasted | | | Trend | | | |
|---|---|---|---|---|---|---|---|
| | **Value** | **High** | **Low** | **Yearly** | **Quarterly** | **Monthly** | **Weekly** |
| 25/6/2021 | 428,974 | 434,384 | 425,410 | 0.08909 | 0.01021 | −0.00046 | 0.0022 |
| 26/6/2021 | 430,002 | 435,584 | 426,229 | 0.08525 | 0.00828 | −0.00129 | 0.00216 |
| 27/6/2021 | 431,030 | 436,785 | 427,048 | 0.08122 | 0.00652 | −0.00163 | −0.0013 |
| 28/6/2021 | 432,058 | 437,986 | 427,867 | 0.07697 | 0.00513 | −0.00131 | −0.00379 |
| 29/6/2021 | 433,086 | 439,187 | 428,686 | 0.07247 | 0.00418 | −0.00062 | −0.00131 |
| 30/6/2021 | 434,114 | 440,388 | 429,505 | 0.06767 | 0.00367 | −0.00003 | 0.0003 |
| 1/7/2021 | 435,142 | 441,589 | 430,324 | 0.06255 | 0.00345 | 0.00017 | 0.00174 |
| 2/7/2021 | 436,170 | 442,790 | 431,143 | 0.05707 | 0.00328 | −0.00001 | 0.0022 |
| 3/7/2021 | 437,198 | 443,991 | 431,962 | 0.05118 | 0.00291 | −0.0004 | 0.00216 |
| 4/7/2021 | 438,226 | 445,192 | 432,781 | 0.04485 | 0.00208 | −0.00075 | −0.0013 |
| 5/7/2021 | 439,254 | 446,393 | 433,600 | 0.03806 | 0.00062 | −0.00082 | −0.00379 |
| 6/7/2021 | 440,282 | 447,594 | 434,419 | 0.03076 | −0.00152 | −0.00052 | −0.00131 |
| 7/7/2021 | 441,310 | 448,795 | 435,238 | 0.02293 | −0.00428 | 0.00005 | 0.0003 |
| 8/7/2021 | 442,338 | 449,996 | 436,057 | 0.01455 | −0.00747 | 0.0006 | 0.00174 |
| 9/7/2021 | 443,366 | 451,197 | 436,876 | 0.00562 | −0.01084 | 0.00089 | 0.0022 |
| 10/7/2021 | 444,394 | 452,397 | 437,695 | −0.00388 | −0.01417 | 0.00093 | 0.00216 |
| 11/7/2021 | 445,422 | 453,598 | 438,514 | −0.01395 | −0.01725 | 0.00086 | −0.0013 |
| 12/7/2021 | 446,450 | 454,799 | 439,333 | −0.02458 | −0.01999 | 0.00077 | −0.00379 |
| 13/7/2021 | 447,478 | 456,000 | 440,152 | −0.03574 | −0.0224 | 0.00062 | −0.00131 |
| 14/7/2021 | 448,506 | 457,201 | 440,971 | −0.04741 | −0.02457 | 0.00039 | 0.0003 |
| 15/7/2021 | 449,534 | 458,402 | 441,790 | −0.05955 | −0.02664 | 0.00022 | 0.00174 |
| 16/7/2021 | 450,562 | 459,603 | 442,609 | −0.07211 | −0.02873 | 0.00027 | 0.0022 |
| 17/7/2021 | 451,590 | 460,804 | 443,428 | −0.08505 | −0.03094 | 0.00051 | 0.00216 |
| 18/7/2021 | 452,618 | 462,005 | 444,247 | −0.09829 | −0.03325 | 0.00065 | −0.0013 |
| 19/7/2021 | 453,646 | 463,206 | 445,066 | −0.11176 | −0.03558 | 0.00043 | −0.00379 |
| 20/7/2021 | 454,674 | 464,407 | 445,885 | −0.12538 | −0.03778 | −0.00002 | −0.00131 |
| 21/7/2021 | 455,702 | 465,608 | 446,704 | −0.13907 | −0.03968 | −0.00027 | 0.0003 |
| 22/7/2021 | 456,730 | 466,809 | 447,523 | −0.15275 | −0.04115 | −0.00007 | 0.00174 |
| 23/7/2021 | 457,758 | 468,010 | 448,342 | −0.16631 | −0.04211 | 0.00035 | 0.0022 |
| 24/7/2021 | 458,786 | 469,211 | 449,161 | −0.17967 | −0.04256 | 0.00044 | 0.00216 |
| 25/7/2021 | 459,814 | 470,411 | 449,980 | −0.19274 | −0.04263 | −0.00008 | −0.0013 |
| 26/7/2021 | 460,842 | 471,612 | 450,799 | −0.20541 | −0.04247 | −0.00095 | −0.00379 |
| 27/7/2021 | 461,870 | 472,813 | 451,618 | −0.21761 | −0.04226 | −0.00157 | −0.00131 |
| 28/7/2021 | 462,898 | 474,014 | 452,437 | −0.22924 | −0.04212 | −0.00152 | 0.0003 |
| 29/7/2021 | 463,926 | 475,215 | 453,256 | −0.24023 | −0.04212 | −0.00094 | 0.00174 |
| 30/7/2021 | 464,954 | 476,416 | 454,075 | −0.25052 | −0.0422 | −0.00026 | 0.0022 |
| 31/7/2021 | 465,982 | 477,617 | 454,894 | −0.26003 | −0.04222 | 0.00014 | 0.00216 |
| 1/8/2021 | 467,010 | 478,818 | 455,713 | −0.26872 | −0.04198 | 0.00011 | −0.0013 |
| 2/8/2021 | 468,039 | 480,019 | 456,532 | −0.27655 | −0.04124 | −0.00022 | −0.00379 |
| 3/8/2021 | 469,067 | 481,220 | 457,351 | −0.2835 | −0.03981 | −0.00062 | −0.00131 |
| 4/8/2021 | 470,095 | 482,421 | 458,171 | −0.28955 | −0.03757 | −0.00084 | 0.0003 |
| 5/8/2021 | 471,123 | 483,622 | 458,990 | −0.29469 | −0.03452 | −0.00069 | 0.00174 |
| 6/8/2021 | 472,151 | 484,823 | 459,809 | −0.29894 | −0.03074 | −0.00021 | 0.0022 |
| 7/8/2021 | 473,179 | 486,024 | 460,628 | −0.30233 | −0.0264 | 0.00038 | 0.00216 |
| 8/8/2021 | 474,207 | 487,224 | 461,447 | −0.30489 | −0.02173 | 0.0008 | −0.0013 |
| 9/8/2021 | 475,235 | 488,425 | 462,266 | −0.30667 | −0.01694 | 0.00094 | −0.00379 |
| 10/8/2021 | 476,263 | 489,626 | 463,085 | −0.30773 | −0.01222 | 0.00089 | −0.00131 |
| 11/8/2021 | 477,291 | 490,827 | 463,904 | −0.30814 | −0.00768 | 0.00081 | 0.0003 |
| 12/8/2021 | 478,319 | 492,028 | 464,723 | −0.30798 | −0.0034 | 0.00069 | 0.00174 |
| 13/8/2021 | 479,347 | 493,229 | 465,542 | −0.30734 | 0.00061 | 0.00049 | 0.0022 |
| 14/8/2021 | 480,375 | 494,430 | 466,361 | −0.30629 | 0.00437 | 0.00027 | 0.00216 |
| 15/8/2021 | 481,403 | 495,631 | 467,180 | −0.30494 | 0.00792 | 0.00021 | −0.0013 |
| 16/8/2021 | 482,431 | 496,832 | 467,999 | −0.30337 | 0.01126 | 0.0004 | −0.00379 |
| 17/8/2021 | 483,459 | 498,033 | 468,818 | −0.30169 | 0.0144 | 0.00063 | −0.00131 |
| 18/8/2021 | 484,487 | 499,234 | 469,637 | −0.29997 | 0.01731 | 0.00057 | 0.0003 |
| 19/8/2021 | 485,515 | 500,435 | 470,456 | −0.29831 | 0.01997 | 0.00018 | 0.00174 |

**Table 8.** *Cont.*

| Date | Forecasted | | | Trend | | | |
|------|-------|------|------|--------|-----------|---------|--------|
| | **Value** | **High** | **Low** | **Yearly** | **Quarterly** | **Monthly** | **Weekly** |
| 20/8/2021 | 486,543 | 501,636 | 471,275 | −0.29679 | 0.02236 | −0.00021 | 0.0022 |
| 21/8/2021 | 487,571 | 502,837 | 472,094 | −0.29549 | 0.02449 | −0.00021 | 0.00216 |
| 22/8/2021 | 488,599 | 504,037 | 472,913 | −0.29445 | 0.02639 | 0.00017 | −0.0013 |
| 23/8/2021 | 489,627 | 505,238 | 473,732 | −0.29374 | 0.0281 | 0.00047 | −0.00379 |
| 24/8/2021 | 490,655 | 506,439 | 474,551 | −0.2934 | 0.02967 | 0.00022 | −0.00131 |
| 25/8/2021 | 491,683 | 507,640 | 475,370 | −0.29346 | 0.03115 | −0.00057 | 0.0003 |
| 26/8/2021 | 492,711 | 508,841 | 476,189 | −0.29394 | 0.03253 | −0.00137 | 0.00174 |
| 27/8/2021 | 493,739 | 510,042 | 477,008 | −0.29484 | 0.0338 | −0.00163 | 0.0022 |
| 28/8/2021 | 494,767 | 511,243 | 477,827 | −0.29616 | 0.03491 | −0.00124 | 0.00216 |
| 29/8/2021 | 495,795 | 512,444 | 478,646 | −0.29788 | 0.03579 | −0.00054 | −0.0013 |
| 30/8/2021 | 496,823 | 513,645 | 479,465 | −0.29996 | 0.0364 | 0.00002 | −0.00379 |
| 31/8/2021 | 497,851 | 514,846 | 480,284 | −0.30237 | 0.03669 | 0.00017 | −0.00131 |

## 6. Discussion and Conclusions

Focusing on the specifics of the ongoing and deadly pandemic, the spread of the disease both epidemiologically and at the level of implementation of preventive and repressive measures is an extremely urgent and important process aimed at revealing the knowledge hidden in the epidemiological data and deciphering indicators that can model the spatio-temporal evolution and spread of the disease.

In this paper, an exploratory study was conducted for the near-real-time analysis of COVID-19 disease data, as well as an intelligent model for predicting disease progression, to assist in deciding on predictive or suppressive measures of social distancing or taking appropriate measures related to the management of the health system. The study was conducted based on an automated system of data collection and analysis, while the medium-term forecast was based on advanced machine learning methods.

The ability to process data in real time, using the tools of intelligent analysis, visualization, and analytical processing, is the basis for methods of dealing with the pandemic and in particular for the effective detection and tracking of active cases. Respectively, the development and use of spatio-temporal forecasts adapted to real data and needs allow the timely methodization of issues related to public health.

Due to the extremely urgent issue, civil protection mechanisms need to incorporate in their technological arsenal systems that are capable of fast to instantaneous data processing, which involve high complexity and possibly great heterogeneity.

Specializing and attempting an evaluation of the results of the forecasting method, it is easy to conclude that the proposed method is a particularly valuable decision support system, as it creates a robust and reliable system of intelligent inference. Reliability is indicative of how the method handles the available data, its mathematical background, and the completeness of the handling of specialized cases that may create noise in the model. In addition, one of the key advantages that need attention is the high reliability that results from the very low error values that resulted from the tests and the forthcoming predictions that were made.

It is also important to note that the proposed methodology models the spread of the disease in the timeliest way, taking into account the actual variation of the recorded cases, which adds complexity to the methodology but also realism. The tests obtained should be considered statistically and semantically significant compared to any other methodology, as they are an indicator of how to study the pandemic at a broader level.

In addition, the proposed model can be used in other scenarios where data are less accurate because Prophet can easily detect the trend of long-term growth with an annual cycle. In addition, the prediction result includes the confidence interval derived from the complete posterior distribution, that is, Prophet provides a data-driven risk estimate. Changepoints (inflection points where the trend changes significantly) can be identified

automatically or defined manually to take more control of forecasting, and the outliers can be handled well by the model itself without any requirement for imputation. In case the forecast is going beyond a certain limit based on case study understanding, it can be fixed by setting up a forecasting cap and modeling using logarithmic growth instead of linear growth. In this study, the time-series data have a natural temporal ordering without taking into account the pandemic waves. The changepoints (the waves of the pandemic) can be identified automatically by Prophet to take more control of forecasting.

Finally, the use of the Prophet algorithm is a very serious proposal for managing chronological data of high complexity and uncertainty such as the one under consideration, which also shows variability, which can be attributed to several unspecified parameters. This technique, as proved mathematically, offers high accuracy predictions and stability, as the overall behavior of the method minimizes noise and at the same time reduces the overall risk of a particularly poor choice that can result from poor sampling or arbitrariness in the parameterization of hyperparameters. The above view is also aided by the fact that the spread of the prediction error is minimized, which clearly states the reliability of the system and the ability to generalize to new data.

Summarizing, we have frequently used Prophet as a replacement for the forecast package in many settings because of two main advantages:

1. Prophet makes it much more straightforward to create a reasonable, accurate forecast. The forecast package includes many different forecasting techniques (ARIMA, exponential smoothing, etc.), each with its own strengths, weaknesses, and tuning parameters. We have found that choosing the wrong model or parameters can often yield poor results, and it is unlikely that even experienced analysts can choose the correct model and parameters efficiently given this array of choices.
2. Prophet forecasts are customizable in ways that are intuitive to non-experts. There are smoothing parameters for seasonality that allow us to adjust how close to fit historical cycles, as well as smoothing parameters for trends that allow us to adjust how aggressively to follow historical trend changes. For growth curves, we can manually specify "capacities" or the upper limit of the growth curve, allowing us to inject our own prior information about how the forecast will grow (or decline). Finally, we can specify irregular holidays to model such as the dates of the local holidays, etc.

However, an important issue at the moment is the fact that in general, modeling a problem with methods such as the proposed one requires a lot of historical data, which is not yet available. However, even if a system based solely on historical data was available, it could only contribute to one aspect of the decisions. A more detailed methodology would be useful in linking technical forecasts to other decision-making factors and study processes that are more complex and potentially more complete. At the same time, no predictions are certain, as the future is seldom repeated in the same way as the past. In addition, it should be noted that forecasts are affected by data reliability and the variables that make up the problem over time. Psychological factors also play an important role in the way people perceive and react to the risk of illness and the fear that it may affect them personally.

Therefore, it is important to keep in mind that these models do not simulate nature itself, which often surprises us, but mathematically represent our perceptions of it and help conditionally explain the epidemiological data, reducing them to a small number of variable factors. In this sense, it is very important to have scientific methodologies and appropriate technical tools or modeling tools such as the proposed one, which can realistically explain similar phenomena and offer valuable assistance in making optimal decisions. It is also important to note that due to the limited knowledge of the new COVID-19 disease, the high level of uncertainty, and the complex socio-political factors influencing the spread of the new virus, no scientifically substantiated methodology for analyzing or predicting the phenomenon is an important legacy. Nevertheless, the ability to accurately predict the course of the pandemic is an extremely difficult and complex task.

Proposals for the development and future improvements of this methodology should focus on further optimizing the parameters of the forecasting system used to achieve an

even more efficient, accurate, and realistic process of approaching the spread of the disease. It would also be important to study the extension of this system by implementing a broader spatio-temporal study at the pan-European or world level to verify the generalization of the method in more complex environments. Finally, an additional element that could be studied in the direction of future expansion concerns the implementation of a hybrid learning system based on the proposed architecture, which with methods of redefining its parameters automatically and in real time can fully automate the forecasting process.

**Author Contributions:** Conceptualization, K.D., D.T. (Dimitrios Taketzis), D.T. (Dimitrios Tsiotas), L.M., L.I., P.K.; methodology, K.D.; software, K.D.; validation, K.D., D.T. (Dimitrios Taketzis), D.T. (Dimitrios Tsiotas), L.M., L.I., P.K.; formal analysis, D.T. (Dimitrios Tsiotas), L.M., L.I.; investigation, K.D.; resources, L.M., L.I., P.K.; data curation, K.D., D.T. (Dimitrios Taketzis), D.T. (Dimitrios Tsiotas), L.M., L.I., P.K.; writing—original draft preparation, K.D., D.T. (Dimitrios Taketzis); writing—review and editing, K.D., D.T. (Dimitrios Taketzis), D.T. (Dimitrios Tsiotas), L.M., L.I., P.K.; visualization, K.D., D.T. (Dimitrios Taketzis), D.T. (Dimitrios Tsiotas); supervision, L.M., L.I., P.K.; project administration, L.M. All authors have read and agreed to the published version of the manuscript.

**Funding:** This research received no external funding.

**Institutional Review Board Statement:** Not applicable.

**Informed Consent Statement:** Not applicable.

**Data Availability Statement:** All data are available freely online at the https://github.com/CSSEGIS andData/COVID-19, access date 10 June 2021.

**Conflicts of Interest:** The authors declare no conflict of interest.

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
