# Peer review of "Pandemic Analytics by Advanced Machine Learning for Improved Decision Making of COVID-19 Crisis"

_processes, doi:10.3390/pr9081267_

Round 1

Reviewer 1 Report

In my opinion, the paper is in general interesting and nice to read. The manuscript deserves to be published only once the authors fix the following issues.

Literature review

  1. The main contributions of the paper are clearly described. Nevertheless, from the current manuscript it is not grasp understanding the novelty of the work. The authors should better highlight the innovative aspects of their work in the manuscript.

Model formulation

  1. A Prophet based model is proposed. The authors should motivate their choice with respect to other models such as SIR based models (e.g., https://doi.org/10.1016/j.arcontrol.2020.09.005, https://doi.org/10.1016/j.arcontrol.2020.10.005, document that could be referenced in the manuscript).
  2. To corroborate the effectiveness of the proposed method, the authors should highlight the advantages with respect to other state-of-the-art alternative techniques.

General

  1. Can the proposed model take restriction measures into account to improve the accuracy of forecast? Moreover, it is well known that testing and screening influence the pandemic dynamics modeling. How the testing (e.g., swabs) affects the prediction achieved by the proposed model?
  2. Does the proposed model work with any waves (first, second, third, etc.)?

Case study

  1. Can the proposed method be used in other scenarios where data are less accurate?

Minor

  1. The authors should check that all the used acronyms are explained. 
  2. Mainly the English is good and there are only a few typos. However the paper should be carefully rechecked.

Author Response

Dear editor and respected reviewers,

We deeply appreciate the time and effort you have spent in reviewing our manuscript entitled "Pandemic Analytics by Advanced Machine Learning for Improved Decision Making of COVID-19 Crisis". Your comments are very helpful for revising and improving our paper. We have revised the manuscript taking into account all the comments exactly to improve the readability of the research paper. We believe these changes have strengthened the rationale and importance of our study.

Cordially

Konstantinos Demertzis, Dimitrios Taketzis, Dimitrios Tsiotas, Lykourgos Magafas, Lazaros Iliadis, Panayotis Kikiras

===========================

Response to Reviewer 1 Comments

In my opinion, the paper is in general interesting and nice to read. The manuscript deserves to be published only once the authors fix the following issues.

Thank you for the careful reading and remarks.

Literature review

Point 1: The main contributions of the paper are clearly described. Nevertheless, from the current manuscript it is not grasp understanding the novelty of the work. The authors should better highlight the innovative aspects of their work in the manuscript.

Response 1: We have revised the manuscript taking into account the comments to improve the novelty of the work. Specifically, we have added in the Introduction the following paragraph that explains things furtherThis paper proposes a novel model for the near-real-time analysis of Covid-19 disease data, as well as an intelligent machine learning system for predicting disease progression, in order to assist in deciding on predictive or suppressive measures of social distancing or taking appropriate measures related to the management of the health system. The proposed system is based on automated data collection and analysis, while the medium-term forecast, based on advanced machine learning methods. Within this context, the proposed method can be applied to different aspects of the COVID-19 temporal spread in Greece and her border countries to present an exploratory study of Covid-19 disease progression (real-time statistics about the cumulative number of infections, deaths, ICU patients, and epidemiological indicators). In practical implementation, the proposed methodology offers an active method for modelling and forecasting the pandemic, capable of removing the disconnected past data from the time-series structure in order to provide a modelling and forecasting tool facilitating decision making and resource management in epidemiology, which can contribute in the on-going fight against the pandemic of COVID-19”. Additional explanations have been added (highlighted by red) in the abstract and conclusion section. The authors thank the reviewer for this comment.

Model formulation

Point 2: A Prophet based model is proposed. The authors should motivate their choice with respect to other models such as SIR based models (e.g., https://doi.org/10.1016/j.arcontrol.2020.09.005, https://doi.org/10.1016/j.arcontrol.2020.10.005, document that could be referenced in the manuscript).

Response 2:  Thank you for this constructive comment. An extensive analysis made in the “2. Related Work” section according to the reviewer’s comment and references suggestion.

Point 3: To corroborate the effectiveness of the proposed method, the authors should highlight the advantages with respect to other state-of-the-art alternative techniques.

Response 3:  The revised manuscript includes the appropriate explanations based on the reviewer’s comments and suggestions, specifically “We have frequently used Prophet as a replacement for the forecast package in many settings because of two main advantages:

  1. Prophet makes it much more straightforward to create a reasonable, accurate forecast. The forecast package includes many different forecasting techniques (ARIMA, exponential smoothing, etc), each with its own strengths, weaknesses, and tuning parameters. We have found that choosing the wrong model or parameters can often yield poor results, and it is unlikely that even experienced analysts can choose the correct model and parameters efficiently given this array of choices.
  2. Prophet forecasts are customizable in ways that are intuitive to non-experts. There are smoothing parameters for seasonality that allow us to adjust how close to fit historical cycles, as well as smoothing parameters for trends that allow us to adjust how aggressively to follow historical trend changes. For growth curves, we can manually specify “capacities” or the upper limit of the growth curve, allowing us to inject our own prior information about how we forecast will grow (or decline). Finally, we can specify irregular holidays to model like the dates of the local holidays, etc”.

General

Can the proposed model take restriction measures into account to improve the accuracy of forecast?

Response 4: Thank you for this comment. In this study, the time-series data have a natural temporal ordering without taking into account the restriction measures. This model is part of a larger and long-term research that is currently being published in stages. After our successful completion of this research, our aim is to create a more sophisticated approach that can be modelling the rationale of the pandemic by creating a time series to a more complex network that takes into account restriction measures, pandemic waves, etc.  

Moreover, it is well known that testing and screening influence the pandemic dynamics modelling. How the testing (e.g., swabs) affects the prediction achieved by the proposed model?

Response 5:  Thank you for this excellent comment that gives us the chance to clarify things further. As we mention previously, the time series data have a natural temporal ordering. This makes time series analysis distinct from cross-sectional studies, in which there is no natural ordering of the observations (e.g. explaining people's infections by reference to their respective test rates levels, where the individuals' data could be entered in any order). The proposed time series analysis is also distinct from spatial data analysis where the observations typically relate to geographical locations (e.g. infection rates by the location as well as the intrinsic characteristics of the tests). The proposed stochastic model for a time series generally reflects the fact that observations close together in time will be more closely related than observations further apart. In addition, the proposed time series model makes use of the natural one-way ordering of time so that values for a given period will be expressed as deriving in some way from past values, rather than from future values (for examble reversibility).

Does the proposed model work with any waves (first, second, third, etc.)?

Response 6:  Thank you for the remarks. In this study, the time-series data have a natural temporal ordering without taking into account the pandemic waves. The changepoints (that are can be the inflection points where the trend changes significantly like the waves of the pandemic) can be identified automatically by Prophet to take more control of forecasting.

Case study

Can the proposed method be used in other scenarios where data are less accurate?

Response 7: The revised manuscript includes the appropriate explanations based on the reviewer’s comments and suggestions. Specifically, …the proposed model can be used in other scenarios where data are less accurate because the Prophet can easily detect the trend of long-term growth with an annual cycle. In addition, the prediction result includes the confidence interval derived from the complete posterior distribution, that is, Prophet provides a data-driven risk estimate. Changepoints (in-flection points where the trend changes significantly) can be identified automatically or defined manually to take more control of forecasting and the outliers can be handled well by the model itself without any requirement for imputation. In case the forecast is going beyond a certain limit based on case study understanding, it can be fixed by setting up a forecasting cap and modelling using logarithmic growth instead of linear growth”. Thank you for this constructive comment.

Minor

The authors should check that all the used acronyms are explained.

Response 8: We have explained all used acronyms. Thank you for the careful reading.

Mainly the English is good and there are only a few typos. However the paper should be carefully rechecked. 

Response 9: We have rearranged the entire paper based on the reviewer’s comments and suggestions. The paper reads much better now, and the work presented has improved to a level acceptable for the readership and the scientific standing of this journal. Thank you for the careful reading.

Reviewer 2 Report

The study you put forward is an important topic that people are concerned about recently. The Covid-19 epidemic prediction and the theme of including government interventions and vaccination to curb the spread of the disease were discussed in detail in the recent study, including various review and study papers. The main focus of this article is to use the proposed method for disease prediction. There are several suggestions for consideration,

  1. Add text to explain the impact of using COVID-19 Government Response Tracker indicators on forecast results.
  2. Supplement the discussion of your proposed Prophet method and other machine learning in prediction-related research and add a summary of opinions.
  3. Abstract can increase the focus of the conclusion.

Author Response

Dear editor and respected reviewers,

We deeply appreciate the time and effort you have spent in reviewing our manuscript entitled "Pandemic Analytics by Advanced Machine Learning for Improved Decision Making of COVID-19 Crisis". Your comments are very helpful for revising and improving our paper. We have revised the manuscript taking into account all the comments exactly to improve the readability of the research paper. We believe these changes have strengthened the rationale and importance of our study.

Cordially

Konstantinos Demertzis, Dimitrios Taketzis, Dimitrios Tsiotas, Lykourgos Magafas, Lazaros Iliadis, Panayotis Kikiras

Response to Reviewer 2 Comments

The study you put forward is an important topic that people are concerned about recently. The Covid-19 epidemic prediction and the theme of including government interventions and vaccination to curb the spread of the disease were discussed in detail in the recent study, including various review and study papers. The main focus of this article is to use the proposed method for disease prediction.

Thank you for the careful reading and remarks.

There are several suggestions for consideration,

  1. Point 1: Add text to explain the impact of using COVID-19 Government Response Tracker indicators on forecast results.

Response 1: We have revised the manuscript taking into account the comments to improve the novelty of the work. Specifically, we have added in the Introduction the following paragraph that explains things furtherThis paper proposes a novel model for the near-real-time analysis of Covid-19 disease data, as well as an intelligent machine learning system for predicting disease progression, in order to assist in deciding on predictive or suppressive measures of social distancing or taking appropriate measures related to the management of the health system. The proposed system is based on automated data collection and analysis, while the medium-term forecast, based on advanced machine learning methods. Within this context, the proposed method can be applied to different aspects of the COVID-19 temporal spread in Greece and her border countries to present an exploratory study of Covid-19 disease progression (real-time statistics about the cumulative number of infections, deaths, ICU patients, and epidemiological indicators). In practical implementation, the proposed methodology offers an active method for modelling and forecasting the pandemic, capable of removing the disconnected past data from the time-series structure in order to provide a modelling and forecasting tool facilitating decision making and resource management in epidemiology, which can contribute in the on-going fight against the pandemic of COVID-19”. Additional explanations have been added (highlighted by red) in the abstract and conclusion section.

On the other hand, the time series data have a natural temporal ordering. This makes time series analysis distinct from cross-sectional studies, in which there is no natural ordering of the observations (e.g. explaining people's infections by reference to their respective test rates levels, where the individuals' data could be entered in any order). The proposed time series analysis is also distinct from spatial data analysis where the observations typically relate to geographical locations (e.g. infection rates by the location as well as the intrinsic characteristics of the tests). The proposed stochastic model for a time series generally reflects the fact that observations close together in time will be more closely related than observations further apart. In addition, the proposed time series model makes use of the natural one-way ordering of time so that values for a given period will be expressed as deriving in some way from past values, rather than from future values (for examble reversibility).

Also, this model is part of a larger and long-term research that is currently being published in stages. After our successful completion of this research, our aim is to create a more sophisticated approach that can be modelling the rationale of the pandemic by creating a time series to a more complex network that takes into account restriction measures, pandemic waves, etc. 

The authors thank the reviewer for this comment.

  1. Point 2: Supplement the discussion of your proposed Prophet method and other machine learning in prediction-related research and add a summary of opinions.

Response 2: The revised manuscript includes the appropriate explanations based on the reviewer’s comments and suggestions, specifically “We have frequently used Prophet as a replacement for the forecast package in many settings because of two main advantages:

  • Prophet makes it much more straightforward to create a reasonable, accurate forecast. The forecast package includes many different forecasting techniques (ARIMA, exponential smoothing, etc), each with its own strengths, weaknesses, and tuning parameters. We have found that choosing the wrong model or parameters can often yield poor results, and it is unlikely that even experienced analysts can choose the correct model and parameters efficiently given this array of choices.
  • Prophet forecasts are customizable in ways that are intuitive to non-experts. There are smoothing parameters for seasonality that allow us to adjust how close to fit historical cycles, as well as smoothing parameters for trends that allow us to adjust how aggressively to follow historical trend changes. For growth curves, we can manually specify “capacities” or the upper limit of the growth curve, allowing us to inject our own prior information about how we forecast will grow (or decline). Finally, we can specify irregular holidays to model like the dates of the local holidays, etc”.

  1. Point 3: Abstract can increase the focus of the conclusion.

Response 3: We have rearranged the entire paper (abstract and conclusion) based on the reviewer’s comments and suggestions. The paper reads much better now, and the work presented has improved to a level acceptable for the readership and the scientific standing of this journal. Thank you for the careful reading.

Round 2

Reviewer 1 Report

Previous comments and concerns have been sufficiently addressed. In the revised paper several improvements have been added.